# DCAD-2000: A Multilingual Dataset across 2000+ Languages with Data Cleaning as Anomaly Detection

Wen Lai[2,3*]  Yingli Shen[1*†]  Shuo Wang[1]  Xueren Zhang[4]
Kangyang Luo[1]  Alexander Fraser[2,3]  Maosong Sun[1,5,6†]

[1]Dept. of Comp. Sci. & Tech., Tsinghua University
[2]Technical University of Munich  [3]Munich Center for Machine Learning
[4]ModelBest Inc.  [5]BNRist Center, Institute for AI, Tsinghua University
[6]Jiangsu Collaborative Innovation Center for Language Ability, Jiangsu Normal University
syl@mail.tsinghua.edu.cn, wen.lai@tum.de

## Abstract

The rapid development of multilingual large language models (LLMs) highlights the need for high-quality, diverse, and well-curated multilingual datasets. In this paper, we introduce DCAD-2000 (Data Cleaning as Anomaly Detection), a large-scale multilingual corpus constructed from newly extracted Common Crawl data and existing multilingual sources. DCAD-2000 covers 2,282 languages, 46.72TB of text, and 8.63 billion documents, spanning 155 high- and medium-resource languages and 159 writing scripts. To overcome the limitations of existing data cleaning approaches, which rely on manually designed heuristic thresholds, we reframe data cleaning as an anomaly detection problem. This dynamic filtering paradigm substantially improves data quality by automatically identifying and removing noisy or anomalous content. By fine-tuning LLMs on DCAD-2000, we demonstrate notable improvements in data quality, robustness of the cleaning pipeline, and downstream performance, particularly for low-resource languages across multiple multilingual benchmarks.

🤗 **Dataset:** `https://huggingface.co/datasets/openbmb/DCAD-2000`
🐙 **Pipeline:** `https://github.com/yl-shen/DCAD-2000`

## 1 Introduction

Large language models (LLMs) have achieved great progress on a variety of NLP tasks by leveraging vast amounts of training data [1]. However, their performance remains heavily biased towards high-resource languages [2, 3]. To improve the multilingual capabilities of LLMs, a common strategy is to incorporate large amounts of non-English data, either by continue pretraining [4] or by instruction tuning in multilingual settings [5]. Therefore, constructing large-scale, high-quality multilingual datasets is crucial for enhancing the multilingual performance of LLMs.

Recent efforts have introduced several large multilingual corpora, including CulturaX [9], HPLT [13], Madlad-400 [10], MaLA [15], and Glotcc [12], which cover 167, 191, 419, 939, and 1,331 languages, respectively. While these datasets have made significant contributions, they exhibit three major limitations, as summarized in Table 1: **(1) Outdated data sources:** These datasets primarily rely on older Common Crawl snapshots[1], which results in outdated knowledge and an elevated risk of

---

*Equal contribution.
†Corresponding author.
[1]`https://commoncrawl.org`

Table 1: Comparison of multilingual datasets constructed from Common Crawl (CC) and our constructed DCAD-2000, focusing on the latest CC version used, the total number of languages supported, distribution across resource categories (high, medium, low, very low), and training readiness. The CC version marked with underline indicates an inferred version due to the lack of explicit specification in the original paper. The "Training-Ready" column indicates whether the dataset is ready for training LLMs without requiring further data cleaning.

| Dataset | CC Version | #Langs (total) | #Langs (high) | #Langs (medium) | #Langs (low) | #Langs (very low) | Training-Ready |
|---|---|---|---|---|---|---|---|
| mC4 [6] | CC-MAIN-2020-34 | 101 | 0 | 43 | 52 | 6 | ✗ |
| OSCAR 23.01 [7] | CC-MAIN-2022-49 | 153 | 6 | 42 | 25 | 80 | ✗ |
| Glot500 [8] | CC-MAIN-2020-34 | 511 | 0 | 108 | 79 | 324 | ✗ |
| CulturaX [9] | CC-MAIN-2022-49 | 167 | 11 | 47 | 27 | 82 | ✗ |
| Madlad-400 [10] | CC-MAIN-2022-33 | 419 | 7 | 46 | 39 | 327 | ✗ |
| MaLA [11] | CC-MAIN-2022-49 | 939 | 1 | 125 | 78 | 735 | ✗ |
| Glotcc [12] | CC-MAIN-2023-50 | 1331 | 0 | 10 | 52 | 1269 | ✗ |
| HPLT-v1.2 [13] | CC-MAIN-2022-40 | 191 | 12 | 53 | 38 | 88 | ✗ |
| Fineweb-2 [14] | CC-MAIN-2024-18 | 1915 | 10 | 62 | 49 | 1794 | ✗ |
| DCAD-2000 | CC-MAIN-2024-46 | 2282 | 13 | 142 | 124 | 2003 | ✓ |

hallucination [16]. **(2) Limited coverage of high- and medium-resource languages**[2]**:** For instance, Fineweb-2 [14], despite supporting 1,915 languages, contains data from only 10 high-resource and 62 medium-resource languages. **(3) Insufficient data cleaning:** Despite being cleaned, recent studies [18, 19] indicate that these datasets still contain a significant amount of noise, which makes them difficult to directly employ in training multilingual LLMs. For example, Sailor [18] reports that 31.11% of Madlad-400 data could still be removed using more advanced cleaning.

Traditional data cleaning workflows [20] often rely on document-level heuristics (e.g., language identification; 21) and fixed thresholds to filter low-quality samples. However, these heuristic thresholds often fail to generalize across languages due to distributional differences in features such as word count, repetition ratios, and perplexity[3]. Notably, while Fineweb-2 fine-tunes thresholds for more than 1,000 languages, this process is computationally intensive and time-consuming.

To address these challenges, we introduce DCAD-2000, a new large-scale, high-quality multilingual dataset that can be directly applied to LLM training. DCAD-2000 covers 2282 languages (155 high/medium languages), incorporating the latest Common Crawl data (November 2024; CC-MAIN-2024-46) and existing multilingual datasets. Additionally, we propose a novel language-agnostic data cleaning approach that treats data cleaning as an anomaly detection [22] problem, distinguishing it from traditional threshold-based methods [14, 23]. Our approach extracts eight statistical features, including *number of words*, *character/word repetition ratio*, *special character/word ratio*, *stopword ratio*, *flagged words ratio*, *language identification score* and *perplexity score*. Anomaly detection algorithms dynamically identify and remove outliers by recognizing deviations from typical document quality metrics.

We conduct a comprehensive analysis of DCAD-2000 with respect to document distribution, linguistic and geographical characteristics, writing scripts, and resource classification (Section 4). By fine-tuning LLMs on DCAD-2000, we validate the effectiveness of its data quality and data cleaning pipeline. Furthermore, we demonstrate the superiority of DCAD-2000 across various language categories (high, medium, low and very low) in multiple multilingual benchmarks, including SIB-200 [24], Glot500 [8] and FLORES-200 [25] (Section 5).

In summary, we make the following contributions:

- We propose a novel data cleaning framework that frames the task as anomaly detection, offering a language-agnostic and adaptive solution without manual threshold tuning.

- We release DCAD-2000, a comprehensive multilingual dataset covering over 2,282 languages, containing 8.63B of documents, 46.72TB of disk size and 159 writing scripts with metadata annotations.

---

[2]We follow the criteria from Flores-101 [17] to categorize languages: High: $> 100M$; Medium: $(1M, 100M)$; Low: $(100K, 1M)$; Very Low: $< 100K$.

[3]Please refer to Appendix A for more details.

- Extensive evaluation across multiple multilingual benchmarks demonstrates the effectiveness of both the data quality and the data cleaning pipeline.

## 2 Related Work

**Multilingual Dataset for Pretraining.** Enhancing the multilingual capabilities of LLMs often involves continuing pretraining on large-scale multilingual datasets [11, 15]. These datasets can be broadly categorized into curated corpora, domain-specific corpora, and web-crawled corpora. **(I) Curated Corpora.** Curated datasets are carefully gathered by experts from high-quality sources such as books [23], academic publications [26], and encyclopedia entries [27, 28, 29]. **(II) Domain-Specific Corpora.** In addition to general-domain data, fine-tuning LLMs on domain-specific datasets is crucial for improving performance in specialized domains like finance [30], healthcare [31], legal [32], and education [33, 34]. **(III) Web-Crawled Corpora.** Web-crawled datasets, particularly those derived from Common Crawl, provide large-scale multilingual coverage by leveraging an open repository of over 250 billion web pages. These datasets include mC4 [6], CC-100 [35], OSCAR [7], Glotcc [12], Fineweb [36], and Fineweb-2 [14]. While curated and domain-specific corpora offer high-quality content with limited language coverage, web-crawled corpora provide broader multilingual coverage but often suffer from noise and lower data quality [18, 19].

**Data Cleaning.** Data cleaning is an essential step in preparing high-quality datasets for training robust LLMs. It involves filtering noisy, irrelevant, or harmful content and can be broadly classified into model-based and heuristic-based approaches [37]. **(I) Model-Based Methods.** Model-based approaches employ classifiers or LLMs to distinguish between high-quality and low-quality data. For instance, content safety models [38] filter out explicit or gambling-related content, while quality classifiers remove low-relevance text [39]. LLM-based methods focus on generating prompts for cleaning [40] or integrating error detection and correction into the pipeline [41, 42]. **(II) Heuristic-Based Methods.** Heuristic approaches apply predefined rules to filter content at both document and sentence levels. At the document level, strategies include filtering by language identification scores or scoring documents with language models [9, 23]. At the sentence level, rules are applied to remove incomplete or irrelevant content, such as HTML tags or excessively short sentences [6, 7]. While model-based methods offer high precision but face scalability challenges, heuristic-based methods are more efficient yet less adaptable to diverse multilingual data.

## 3 DCAD-2000

To overcome the limitations of existing multilingual datasets, we introduce DCAD-2000, a large-scale, high-quality multilingual dataset constructed by integrating data from latest version of Common Crawl and existing multilingual datasets (Section 3.1). This dataset is cleaned using our proposed novel framework, which treats data cleaning as an anomaly detection problem (Section 3.2). The construction of DCAD-2000 is supported by robust computational resources, as detailed in Section 3.3.

### 3.1 Data Collection

To ensure comprehensive and robust multilingual data representation, DCAD-2000 integrates data from four main sources: MaLA, Fineweb, Fineweb-2, and newly extracted Common Crawl data. Each source is selected based on its unique contribution to multilingual coverage, data quality, and freshness, with careful consideration to complementarity to minimize redundancy. Specifically, MaLA and Fineweb-2 are prioritized due to their broad language coverage and high-quality curation, which complements other widely used datasets like mC4 [6] and OSCAR [7].

**MaLA Corpus [11].** The MaLA corpus covers 939 languages, aggregating data from diverse sources including Bloom [43], CC100 [35], Glot500 [8], among others. Deduplication is performed using MinHashLSH [44], which is particularly effective in removing near-duplicate entries that often arise from common web sources. Language codes are based on ISO 639-3[4] standards, and language-specific scripts are supported by GlotScript[5].

---

[4] https://en.wikipedia.org/wiki/ISO_639-3
[5] https://github.com/cisnlp/GlotScript

**Fineweb Corpus [36].** Fineweb is a high-quality English web dataset extracted from Common Crawl, consisting of over 15 trillion tokens and updated monthly. Data cleaning and deduplication are performed using the Datatrove library.[6] For DCAD-2000, we incorporate data from the November 2024 release (`CC-MAIN-2024-46`) to ensure freshness and up-to-date relevance of the data.

**Fineweb-2 Corpus [14].** Fineweb-2 expands Fineweb to include multilingual data, covering 1,915 languages. It processes 96 Common Crawl dumps from 2013 (`CC-MAIN-2013-20`) to April 2024 (`CC-MAIN-2024-20`). The deduplication process within Fineweb-2 is similarly handled using the Datatrove library, ensuring the exclusion of redundant entries and maintaining high-quality multilingual coverage.

**Newly Extracted Common Crawl Data.** To incorporate the most recent multilingual data, we extract and process Common Crawl dumps from May 2024 (`CC-MAIN-2024-22`) to November 2024 (`CC-MAIN-2024-46`). Using the Fineweb-2 pipeline[7], we process 21.54TB of multilingual data, ensuring that the data remains fresh and suitable for downstream tasks. This further extends the multilingual data pool and enhances the coverage across underrepresented languages.

## 3.2 Data Cleaning as Anomaly Detection

Traditional data cleaning methods rely on fixed thresholds for document-level features, making them less adaptable to the diversity of multilingual data. To address this, we propose a novel framework that formulates data cleaning as an anomaly detection task, which involves feature extraction (Section 3.2.1) and anomaly detection (Section 3.2.2).

### 3.2.1 Feature Extraction

Inspired by Roots [23] and CulturaX [9], we extract eight statistical features from each document to evaluate text quality. Each feature is selected for its ability to capture important characteristics of the text, contributing to robust anomaly detection. Let $t$ represent a document; the extracted features are:

- **Number of Words,** $n_w(t)$**:** Total number of tokens after language-specific tokenization, providing a coarse measure of document length and helping identify extremely short or excessively long outliers.
- **Character Repetition Ratio,** $r_c(t)$**:** Fraction of repeated character sequences (e.g., "aaaaa" or "!!!!!"), which often signal encoding artifacts, copy-paste errors, or spam-like content.
- **Word Repetition Ratio,** $r_w(t)$**:** Proportion of repeated lexical items, useful for detecting low-information documents that exhibit looping or template-like patterns.
- **Special Characters Ratio,** $r_s(t)$**:** Fraction of characters belonging to special symbol categories. We employ the curated language-specific symbol lists provided in the ROOTs Corpus [23], covering punctuation, numeric symbols, whitespace variants, and emojis. A high $r_s(t)$ may indicate adversarial inputs or unstructured noise.
- **Stopwords Ratio,** $r_{\mathbf{stop}}(t)$**:** Ratio of stopwords derived from Fineweb-2's multilingual stopword lexicons. This metric captures the functional-to-content word balance, offering a lightweight approximation of linguistic naturalness.
- **Flagged Words Ratio,** $r_{\mathbf{flag}}(t)$**:** Fraction of tokens that appear in curated lists of toxic or profane vocabulary such as *Toxicity-200* [25] and community-maintained sources[8]. This feature enables early detection of harmful or sensitive content.
- **Language Identification (LID) Score,** $s_{\mathbf{lid}}(t)$**:** Confidence score produced by GlotLID [21], a language identifier supporting over 2,000 languages. Lower scores may indicate code-switching, mislabeling, or mixed-script anomalies.
- **Perplexity Score,** $s_{\mathbf{ppl}}(t)$**:** We compute a language model perplexity score using KenLM [45] models trained per language on the November 2023 snapshot of multilingual Wikipedia[9]. This feature provides a lightweight proxy for linguistic fluency.

---

[6]`https://github.com/huggingface/datatrove`

[7]`https://github.com/huggingface/fineweb-2`

[8]`https://github.com/thisandagain/washyourmouthoutwithsoap`

[9]KenLM models are only trained for languages with sufficient clean Wikipedia data (minimum 10,000 high-quality sentences). For other languages, we assign a default perplexity score of 500.

The feature vector for each document is defined as:
$$\mathbf{x} = [n_w(t), \, r_c(t), \, r_w(t), \, r_s(t), \, r_{\text{stop}}(t), \, r_{\text{flag}}(t), \, s_{\text{lid}}(t), \, s_{\text{ppl}}(t)]^\top \in \mathbb{R}^8. \tag{1}$$

### 3.2.2 Anomaly Detection

After extracting feature vectors $\mathbf{x} \in \mathbb{R}^8$, we standardize each feature to handle differences in scale. The standardized value $\tilde{x}_j$ for the $j$-th feature is given by:
$$\tilde{x}_j = \frac{x_j - \mu_j}{\sigma_j}, \quad j = 1, \ldots, 8. \tag{2}$$

where $\mu_j$ and $\sigma_j$ are the mean and standard deviation of the $j$-th feature across the dataset. The standardized feature vector is:
$$\tilde{\mathbf{x}} = \frac{\mathbf{x} - \boldsymbol{\mu}}{\boldsymbol{\sigma}}. \tag{3}$$

where $\boldsymbol{\mu} = [\mu_1, \mu_2, \ldots, \mu_8]^\top$ and $\boldsymbol{\sigma} = [\sigma_1, \sigma_2, \ldots, \sigma_8]^\top$ are the vectors of means and standard deviations, respectively.

Take Isolation Forest [46] as an example[10], we compute an anomaly score $\phi(\tilde{\mathbf{x}})$ for each document. The Isolation Forest algorithm assigns anomaly scores based on the average path length required to isolate a data point in a decision tree. Specifically, for a document represented by $\tilde{\mathbf{x}}$, the anomaly score is defined as:
$$\phi(\tilde{\mathbf{x}}) = 2^{-\frac{h(\tilde{\mathbf{x}})}{c(n)}}. \tag{4}$$

where $h(\tilde{\mathbf{x}})$ is the average path length for $\tilde{\mathbf{x}}$ across all trees in the Isolation Forest, and $c(n)$ is the average path length of a point in a binary tree with $n$ samples, given by:
$$c(n) = 2H(n-1) - \frac{2(n-1)}{n}. \tag{5}$$

where $H(i)$ is the $i$-th harmonic number, defined as $H(i) = \sum_{k=1}^{i} \frac{1}{k}$.

An anomaly score $\phi(\tilde{\mathbf{x}}) : \mathbb{R}^8 \to \mathbb{R}$ is defined to quantify how far a document deviates from typical data. Higher scores indicate a higher likelihood of anomalies. To classify a document, we use the decision rule:
$$f(\tilde{\mathbf{x}}) = \begin{cases} 1, & \text{if } \phi(\tilde{\mathbf{x}}) < \tau, \\ -1, & \text{if } \phi(\tilde{\mathbf{x}}) \geq \tau. \end{cases} \tag{6}$$

where $\tau \in \mathbb{R}$ is a hyperparameter determined empirically or through cross-validation.[11]

Once the anomaly scores $\phi(\tilde{\mathbf{x}})$ are computed for all samples in the standardized dataset $\tilde{\mathcal{X}} = \{\tilde{\mathbf{x}}_1, \ldots, \tilde{\mathbf{x}}_N\}$, we partition the dataset into two subsets:
$$\mathcal{X}_{\text{keep}} = \{\tilde{\mathbf{x}} \in \tilde{\mathcal{X}} : f(\tilde{\mathbf{x}}) = 1\}, \tag{7}$$
$$\mathcal{X}_{\text{remove}} = \{\tilde{\mathbf{x}} \in \tilde{\mathcal{X}} : f(\tilde{\mathbf{x}}) = -1\}. \tag{8}$$

Following anomaly detection, the dataset is partitioned into a clean subset $\mathcal{X}_{\text{keep}}$ and an anomalous subset $\mathcal{X}_{\text{remove}}$. The former is retained for downstream tasks such as model training, while the latter may be discarded or further examined for potential data quality issues.

### 3.2.3 Visualization

To qualitatively evaluate the separation achieved by our data cleaning framework, we present scatter plots of the eight feature dimensions in Figure 1, with data points color-coded by their anomaly labels. These visualizations facilitate the interpretation of decision boundaries and highlight the features that contribute most significantly to the detection process. We observe well-defined clusters separating anomalous and non-anomalous data points, with anomalies exhibiting distinct patterns compared to the majority of the data. Features such as the language identification score ($s_{\text{lid}}(t)$) and perplexity score ($s_{\text{ppl}}(t)$) are expected to be particularly discriminative in identifying anomalies, as they capture linguistic irregularities and unexpected text patterns. For example, low $lid$ or unusually high $ppl$ scores often indicate problematic text, such as spam, low-quality content, or noise. The framework effectively identifies and removes such low-quality text samples, which can be easily visualized by the separation of these points in the scatter plots.

---

[10]We also evaluate some other algorithms, please refer to Section 5 for more details.

[11]We use the default settings of the specific anomaly detection algorithm in Scikit-learn, applying these settings globally rather than individually for each feature or language.

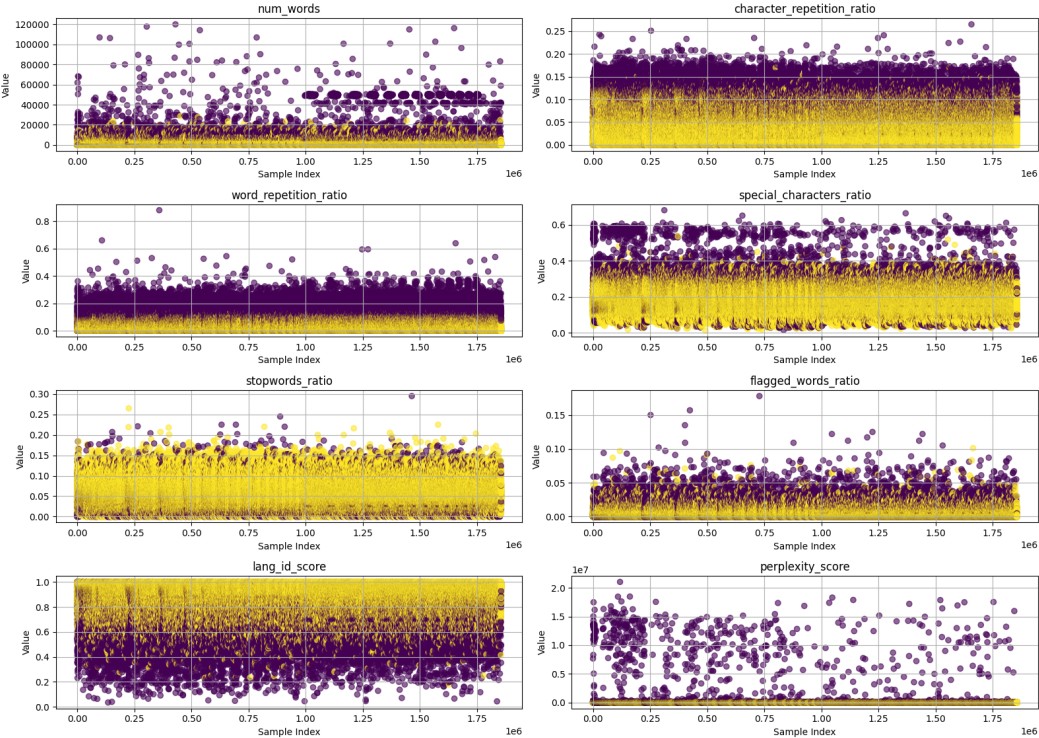

Figure 1: Scatter plots of eight features extracted from a Chinese corpus during the data cleaning process, with data points color-coded according to their anomaly labels. The yellow points represent high-quality data, while the purple points indicate low-quality data.

## 3.3 Computational Resources

The construction of the DCAD-2000 dataset leveraged Ksyun servers[12] to process and clean the multilingual data efficiently. Each server instance is equipped with 32 CPU cores, 128GB of memory, and 100GB of disk storage, which is utilized for intermediate data handling and memory-intensive operations such as anomaly detection. The workload is managed using container orchestration tools, Kubernetes[13], with up to 100 parallel tasks running per job to ensure scalability.

## 4 Dataset Analysis

In this section, we analyze the characteristics of DCAD-2000, focusing on document distribution across sources, geographic and script coverage, resource categorization of languages, and the effect of data cleaning on dataset size and quality.

**Document Distribution Across Data Sources.** The DCAD-2000 dataset is derived from four primary sources: MaLA, Fineweb, Fineweb-2, and Newly Extracted Common Crawl data (New CC), as described in Section 3.1. Figure 2a presents the distribution of documents across these sources, with Fineweb-2 and New CC collectively contributing 47.5% and 39.3% of the total dataset, respectively. These two sources play a significant role in ensuring the dataset's emphasis on both language diversity (Fineweb-2) and corpus freshness (New CC). MaLA, though contributing 11.1% of the total dataset, brings in valuable content from non-Common Crawl sources, further enriching the diversity of the dataset, especially for low-resource languages.

---

[12]https://www.ksyun.com
[13]https://kubernetes.io

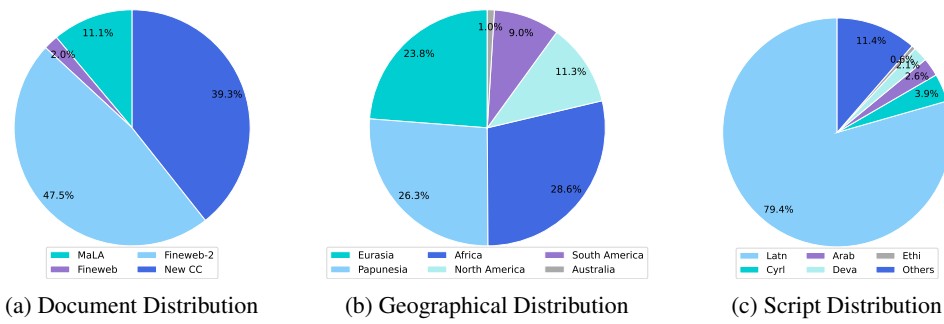

(a) Document Distribution     (b) Geographical Distribution     (c) Script Distribution

Figure 2: Document distribution and linguistic diversity in DCAD-2000.

**Geographical Coverage of Languages.** Figure 2b shows the geographical distribution of languages in DCAD-2000, based on the number of unique languages available in each region, as classified by Glottolog.[14] The dataset spans languages from all major world regions, with the largest proportions originating from Africa (28.6%), Papunesia (26.3%) and Eurasia (23.8%). This coverage ensures robust support for multilingual applications across varied regional contexts, including densely populated areas like Eurasia and sparsely populated regions such as Papunesia and Australia. While Eurasia is more heavily represented, this diversity of linguistic coverage helps ensure that the dataset remains useful for training LLMs in diverse regional environments.

**Script Distribution.** Figure 2c illustrates the distribution of languages in DCAD-2000 by writing system. The dataset supports 159 scripts, with the Latin script dominating at 79.4%, followed by Cyrillic (3.9%), Arabic (2.6%), and Devanagari (2.1%), among others. This diversity in scripts enables a wide range of cross-lingual and script-specific tasks. However, the inclusion of minority scripts, especially those with limited resources, poses unique challenges, such as optical character recognition (OCR) difficulties for certain scripts or inconsistent text quality. Despite these challenges, DCAD-2000 ensures comprehensive coverage by including data from diverse scripts. A complete list of supported scripts is provided in Appendix B.

**Language Resource Classification.** Following the classification approach proposed by Flores [17], we categorize languages in DCAD-2000 into four groups based on corpus size: high-resource, medium-resource, low-resource, and extremely low-resource. Table 1 shows the distribution across these categories. The dataset includes 155 high- and medium-resource languages, while low-resource languages make up a significant portion, which reflects DCAD-2000's commitment to supporting underrepresented languages. Notably, DCAD-2000 surpasses other corpora in its balance between high-resource and low-resource languages, which can have a significant impact on multilingual model training. The distribution of languages across categories ensures that the dataset is well-suited for developing models that perform effectively across diverse language resources.

**Impact of Data Cleaning.** We summarize the document count, token count, and disk size of the high/medium/low resource languages in DCAD-2000 before and after the data cleaning process. Complete details are provided in Appendix C. The cleaning process results in the removal of a substantial amount of noisy data, even from datasets like MaLA, Fineweb, and Fineweb-2, which had already been subject some cleaning. This aligns with the findings from [18, 19]. For example, in the MaLA dataset, 8.05 million documents are removed for the *hbs_Latn* language, which suggests the necessity of rigorous data cleaning to enhance dataset quality. Overall, the cleaning process removed approximately 7.69% of the documents across all languages, significantly improving the quality of the dataset by reducing noise and increasing relevance for model training (Section 5).

# 5 Evaluation

Following Fineweb-2 [14], we conduct a series of experiments on the FineTask benchmark[15] to evaluate the effectiveness of our proposed data cleaning pipeline and assess the quality of the DCAD-2000 dataset. FineTask comprises tasks in nine languages (i.e., *Chinese*, *French*, *Arabic*, *Russian*, *Thai*,

---

[14]Geographic data source: `https://glottolog.org`
[15]`https://huggingface.co/spaces/HuggingFaceFW/blogpost-fine-tasks`

Table 2: The performance of various anomaly detection algorithms. **Bold** and underlined numbers indicates the best and second-best results respectively.

| | LLaMA-3.2-1B | | | | | Qwen-2.5-7B | | | | | Aya-expanse-32B | | | | |
|---|---|---|---|---|---|---|---|---|---|---|---|---|---|---|---|
| | **Baseline** | **Iso_Forest** | **OC_SVM** | **LOF** | **K-Means** | **Baseline** | **Iso_Forest** | **OC_SVM** | **LOF** | **K-Means** | **Baseline** | **Iso_Forest** | **OC_SVM** | **LOF** | **K-Means** |
| Arabic | 0.07 | **0.21** | 0.18 | **0.21** | 0.14 | 0.63 | **0.71** | 0.68 | 0.65 | 0.68 | 0.69 | **0.75** | 0.70 | 0.71 | 0.69 |
| Turkish | 0.07 | 0.27 | **0.29** | 0.17 | 0.15 | 0.65 | 0.72 | **0.73** | 0.67 | 0.68 | 0.75 | **0.79** | 0.77 | 0.76 | 0.77 |
| Swahili | 0.08 | **0.29** | 0.25 | 0.19 | 0.19 | 0.25 | 0.34 | 0.27 | **0.35** | 0.27 | 0.35 | **0.44** | 0.36 | 0.37 | 0.41 |
| Russian | 0.10 | **0.24** | 0.19 | 0.18 | 0.15 | 0.74 | **0.79** | 0.75 | 0.75 | 0.76 | 0.77 | **0.82** | 0.79 | 0.80 | 0.79 |
| Telugu | 0.02 | **0.06** | 0.05 | 0.04 | 0.04 | 0.16 | 0.24 | **0.26** | 0.20 | 0.21 | 0.15 | 0.25 | 0.19 | 0.21 | **0.27** |
| Thai | 0.14 | **0.21** | 0.18 | 0.18 | 0.15 | 0.57 | **0.64** | 0.59 | 0.59 | 0.61 | 0.38 | **0.46** | 0.42 | 0.43 | 0.40 |
| Chinese | 0.12 | **0.32** | 0.28 | 0.25 | 0.21 | 0.75 | **0.82** | 0.77 | 0.76 | 0.78 | 0.69 | **0.75** | 0.71 | 0.71 | 0.73 |
| French | 0.11 | 0.35 | **0.37** | 0.30 | 0.23 | 0.74 | **0.80** | 0.76 | 0.76 | 0.75 | 0.74 | **0.79** | 0.76 | 0.76 | 0.76 |
| Hindi | 0.07 | **0.21** | 0.17 | 0.16 | 0.14 | 0.49 | **0.57** | 0.52 | 0.53 | 0.52 | 0.69 | **0.74** | 0.72 | 0.73 | 0.72 |

*Hindi*, *Turkish*, *Swahili*, and *Telugu*), and covers a diverse set of NLP tasks, including reading comprehension, commonsense reasoning, natural language understanding, and text generation. To investigate the impact of different data cleaning strategies and anomaly detection algorithms, we continue pretraining on three typical LLMs: `LLaMA-3.2-1B`[47], `Qwen-2.5-7B`[48], and `Aya-expanse-32B`[49]. Additionally, we analyze the performance across different resource categories using the SIB-200[24], Glot500-c [8], and FLORES-200 [25] benchmarks. We report normalized accuracy for FineTask, raw accuracy for SIB-200, negative log-likelihood (NLL) for Glot500-c, and BLEU scores for FLORES-200. Full experimental settings and results are provided in Appendix D.

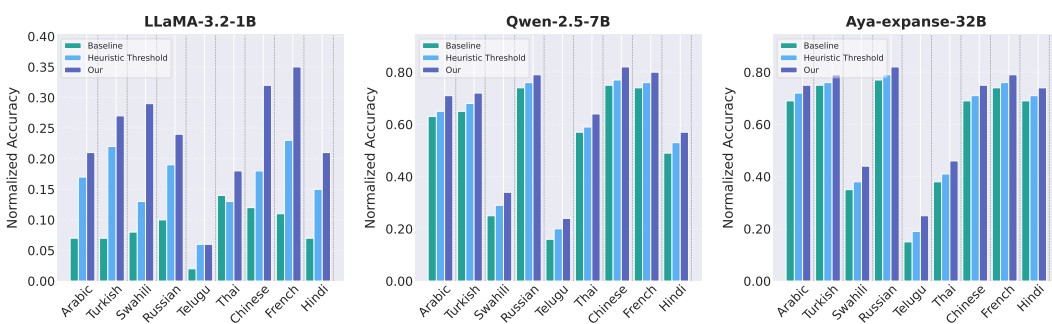

Figure 3: The performance comparison of models trained using various data cleaning methods.

**Impact of Different Data Cleaning Strategies.** We evaluate the effectiveness of our proposed anomaly detection-based data cleaning framework by comparing model performance across various cleaning strategies. As illustrated in Figure 3, the baseline model trained on raw, unfiltered data consistently underperforms relative to all cleaning methods. This performance gap is primarily due to noisy, irrelevant, or inconsistent data that hinders model generalization. Traditional threshold-based filtering[16], which removes low-quality samples using fixed rules based on features, yields modest improvements. In contrast, our anomaly detection-based approach dynamically identifies and filters anomalous or noisy data, resulting in significantly enhanced model performance. Models trained using our method achieve normalized accuracy improvements of approximately 5–20% over the baseline, and outperform the threshold-based approach by 3–10%. Threshold-based approaches trade accuracy for efficiency, whereas our framework, despite higher computational demands, uncovers subtle and cross-lingual data anomalies that fixed rules frequently overlook.

**Comparison of Anomaly Detection Algorithms.** We compare several classical anomaly detection algorithms to identify the most effective approach for constructing DCAD-2000. The evaluated methods include Isolation Forest (ISO_Forest;46), One-Class SVM (OC_SVM; 50), Local Outlier Factor (LOF;51), and K-Means [52], using implementations from scikit-learn[17]. We provide the comparison of different algorithms in Appendix D.3. Table 2 reports the performance of these algorithms in cleaning the dataset. While all anomaly detection methods outperform the unfiltered baseline, the performance of OC_SVM, LOF, and K-Means is notably inconsistent. These algorithms often require extensive parameter tuning (e.g., selecting the number of neighbors for LOF or the kernel type for OC_SVM), which introduces sensitivity to hyperparameters and increases computational overhead. In contrast, ISO_Forest demonstrates more stable and robust performance across experiments, attributed to its efficiency in handling noisy, high-dimensional multilingual data. Unlike other methods,

---
[16]We use the implementation from `https://github.com/bigscience-workshop/data-preparation`
[17]`https://scikit-learn.org`

Table 3: Performance across different language categories. We use accuracy (↑) in SIB-200, negative log-likelihood (↓) in Glot500-c and BLEU (↑) in FLORES-200. Improvements are highlighted accordingly.

| | | LLaMA-3.2-1B | | | Qwen-2.5-7B | | | Aya-expanse-32B | | |
|---|---|---|---|---|---|---|---|---|---|---|
| | | Fineweb-2 | New CC | DCAD-200 | Fineweb-2 | New CC | DCAD-200 | Fineweb-2 | New CC | DCAD-200 |
| **SIB-200 (↑)** | H | 8.24 | 8.86 | **10.37** $\uparrow_{2.13}$ | 33.41 | 34.53 | **38.26** $\uparrow_{4.85}$ | 41.72 | 42.41 | **47.93** $\uparrow_{6.21}$ |
| | M | 7.31 | 7.92 | **9.15** $\uparrow_{1.84}$ | 28.72 | 29.86 | **32.65** $\uparrow_{3.93}$ | 32.25 | 33.39 | **38.16** $\uparrow_{5.91}$ |
| | L | 6.06 | 6.45 | **7.83** $\uparrow_{1.77}$ | 23.58 | 24.22 | **27.12** $\uparrow_{3.54}$ | 26.87 | 27.57 | **33.24** $\uparrow_{6.37}$ |
| | VL | 3.68 | 4.27 | **5.24** $\uparrow_{1.56}$ | 13.25 | 15.43 | **21.57** $\uparrow_{8.32}$ | 17.23 | 19.5 | **26.38** $\uparrow_{9.15}$ |
| **Glot500-c test (↓)** | H | 426.37 | 403.58 | **373.14** $\downarrow_{53.23}$ | 347.21 | 334.18 | **303.38** $\downarrow_{43.83}$ | 273.85 | 257.24 | **225.28** $\downarrow_{48.57}$ |
| | M | 446.28 | 436.94 | **423.75** $\downarrow_{22.53}$ | 385.72 | 389.24 | **369.15** $\downarrow_{16.57}$ | 326.92 | 321.16 | **302.53** $\downarrow_{24.39}$ |
| | L | 503.38 | 493.27 | **473.96** $\downarrow_{29.42}$ | 426.33 | 419.25 | **404.28** $\downarrow_{22.05}$ | 372.62 | 367.26 | **341.34** $\downarrow_{31.28}$ |
| | VL | 584.55 | 569.34 | **532.86** $\downarrow_{51.69}$ | 479.04 | 463.36 | **433.48** $\downarrow_{45.56}$ | 396.33 | 392.33 | **385.86** $\downarrow_{10.47}$ |
| **FLORES-200 (↑)** (Eng–X) | H | 3.14 | 3.82 | **5.26** $\uparrow_{2.12}$ | 15.24 | 16.07 | **18.47** $\uparrow_{3.23}$ | 23.45 | 24.33 | **26.33** $\uparrow_{2.88}$ |
| | M | 2.75 | 2.94 | **3.89** $\uparrow_{1.14}$ | 12.83 | 13.46 | **15.49** $\uparrow_{2.66}$ | 19.36 | 20.21 | **21.62** $\uparrow_{2.26}$ |
| | L | 2.27 | 2.41 | **3.14** $\uparrow_{0.87}$ | 8.94 | 9.28 | **10.25** $\uparrow_{1.31}$ | 16.61 | 17.24 | **18.36** $\uparrow_{1.75}$ |
| | VL | 1.85 | 2.05 | **2.35** $\uparrow_{0.50}$ | 6.33 | 7.25 | **9.05** $\uparrow_{2.72}$ | 12.51 | 13.16 | **14.77** $\uparrow_{2.26}$ |
| **FLORES-200 (↑)** (X–Eng) | H | 3.94 | 3.98 | **4.26** $\uparrow_{0.32}$ | 16.31 | 16.92 | **18.84** $\uparrow_{2.53}$ | 23.86 | 24.13 | **26.94** $\uparrow_{3.08}$ |
| | M | 3.52 | 3.66 | **3.80** $\uparrow_{0.28}$ | 13.65 | 14.05 | **16.27** $\uparrow_{2.62}$ | 20.45 | 20.36 | **22.53** $\uparrow_{2.17}$ |
| | L | 3.05 | 3.12 | **3.24** $\uparrow_{0.19}$ | 9.47 | 10.22 | **11.48** $\uparrow_{2.01}$ | 17.67 | 17.82 | **18.93** $\uparrow_{1.26}$ |
| | VL | 2.73 | 2.83 | **3.14** $\uparrow_{0.41}$ | 7.28 | 7.81 | **9.65** $\uparrow_{2.37}$ | 13.25 | 13.56 | **15.88** $\uparrow_{2.63}$ |

ISO_Forest delivers reliable results without intensive hyperparameter tuning, making it particularly suitable for large-scale multilingual datasets. However, ISO_Forest can be more computationally demanding than simpler methods like K-Means, especially in high-dimensional settings (our feature vectors have eight dimensions, as described in Section 3.2.2). Despite this trade-off, its robustness and scalability establish ISO_Forest as the most appropriate choice for data cleaning in DCAD-2000.

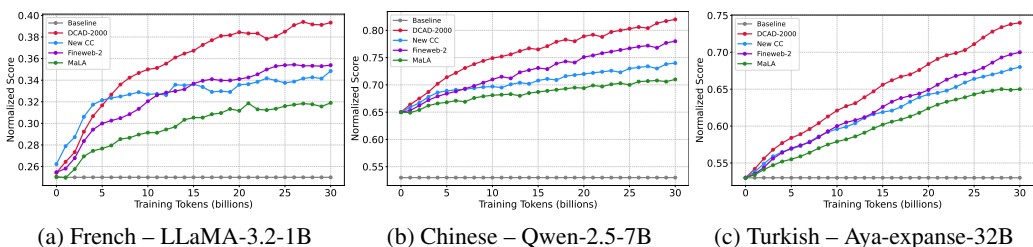

(a) French – LLaMA-3.2-1B    (b) Chinese – Qwen-2.5-7B    (c) Turkish – Aya-expanse-32B

Figure 4: Comparison of DCAD-2000 with existing multilingual corpora for three languages—French, Chinese, and Turkish—evaluated using different multilingual LLMs.

**Comparison with Other Multilingual Datasets.** To validate the quality of DCAD-2000, we compare it against existing multilingual corpora on the FineTask benchmark. These corpora include datasets constructed from *New CC*, *MaLA*, and *Fineweb-2* as described in Section 3.1. As shown in Figure 4, models trained on DCAD-2000 consistently outperform those trained on other datasets, achieving higher normalized accuracy. The improvements can be attributed to the enhanced data quality, diversity, and reduced noise resulting from our comprehensive cleaning pipeline. Specifically, DCAD-2000 provides greater linguistic diversity and a more balanced representation of low-resource languages, leading to improved performance on tasks involving underrepresented languages like Swahili and Telugu.

**Analysis by the Categories of Language Resources.** Table 3 presents model performance across languages categorized by resource levels (High, Medium, Low, and Very Low). Across all benchmarks and model sizes, DCAD-2000 consistently outperforms Fineweb-2 and New CC. While the gains are modest for high-resource languages, improvements are substantial for low- and very low-resource languages, reaching up to +9.15 accuracy on SIB-200 and −53.23 NLL on Glot500-c, which highlights the effectiveness of our cleaning pipeline in improving data quality where it is most needed. The BLEU results on FLORES-200 further validate these trends, with notable improvements in both English-to-X and X-to-English translation tasks. These consistent gains across tasks and languages demonstrate that DCAD-2000 enables more balanced multilingual performance and is well-suited for training inclusive, high-quality language models.

**Manual Quality Evaluation of Cleaning Pipeline.** To assess the effectiveness of our cleaning pipeline, we conduct a manual quality evaluation on five representative languages: English, Chinese, German, Japanese, and French. More specifically, we randomly sampled 100 retained and 100

deleted documents per language, with each document labeled by a proficient annotator as "*Good*," "*Borderline*," or "*Bad*." The evaluation revealed that the pipeline retained high-quality content with minimal residual noise ( 4.4%) and low false positive rates ( 5.2%). These results confirm the robustness of our unsupervised, anomaly-detection-based method in effectively removing low-quality content while preserving valuable data. Full details of the experimental setup and results can be found in the Appendix E.

**Further Investigation.** To evaluate the practical trade-offs between conventional heuristic filtering and our anomaly-based framework, we conduct a controlled cost–benefit analysis and found that DCAD incurs only minor computational overhead while improving downstream task performance; please refer to Appendix F for more details. To assess the robustness of different feature combinations, we performed an ablation study on the 8-dimensional feature vector and observed that each feature contributes meaningfully, with the Language Identification confidence score being particularly critical; please refer to Appendix G for more details. To justify the practical choice of anomaly detector and explore future extensions, we analyzed the trade-offs between classical and modern deep anomaly detection methods and highlighted the scalability, interpretability, and resource efficiency of Isolation Forest; please refer to Appendix H for more details.

# 6    Conclusion

In this paper, we introduce DCAD-2000, a large-scale multilingual dataset designed to address the increasing demand for high-quality and diverse training data for multilingual LLMs. Our dataset spans 2,282 languages, providing comprehensive coverage across various geographic regions, scripts (159 scripts), and larger coverage of high/medium resource languages (155 languages). To avoid manually setting thresholds during the data cleaning process, we propose a novel framework that reframes data cleaning as an anomaly detection task. This dynamic approach ensures effective identification and removal of anomalous data from noisy datasets. Empirical experiments demonstrate the effectiveness of our proposed data cleaning framework and the high quality of the DCAD-2000 dataset across multiple multilingual benchmarks.

# 7    Limitations

This work has the following limitations: **(i)** Although the proportion of high/medium/low resource languages in DCAD-2000 has greatly increased compare to existing multilingual datasets, a significant portion of the languages are still very low resource languages. Future work will explore to collect data for extremely low-resource languages through other modalities (e.g., images) through technologies like OCR. **(ii)** We evaluate the new data cleaning framework only on four classical anomaly detection algorithms; however, since the framework is algorithm-independent, it should also be effective with other anomaly detection algorithms. **(iii)** For language identification, we use GlotLID [21], a FastText-based model whose limitations in handling massive multilinguality have been discussed in previous works [53]. However, since the data cleaning pipeline is language-agnostic, other language identification models can also be employed. **(iv)** We use a classical, feature-based anomaly detection algorithm rather than modern deep or embedding-based methods [54] because of the lack of clean reference distributions, the need for scalability across thousands of languages, and resource constraints. We will explore incorporating semantic embedding-based or lightweight deep anomaly detectors in future work to capture subtler anomalies that our current approach may miss.

# Acknowledgments

This work is supported by the Beijing Municipal Science and Technology Plan Project (Z241100001324025) and is also supported by the AI9Stars community. In addition, support was received from the European Research Council (ERC) under the Horizon Europe Research and Innovation Program of the European Union (Grant Agreement No. 101113091), as well as from the German Research Foundation (DFG; Grant FR 2829/7-1).

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

# A  Statistical Analysis of Multilingual Datasets

In this section, we explore the statistical characteristics of the dataset through visual analysis, focusing on the distribution of data across different languages and the variations observed across different shards. We highlight the limitations of existing data cleaning methods that rely on fixed thresholds, particularly in the imbalanced data distribution scenarios. Specifically, when there are substantial discrepancies in word count distributions, these threshold-based cleaning methods are prone to errors, which fail to accurately distinguish between high-quality and low-quality data.

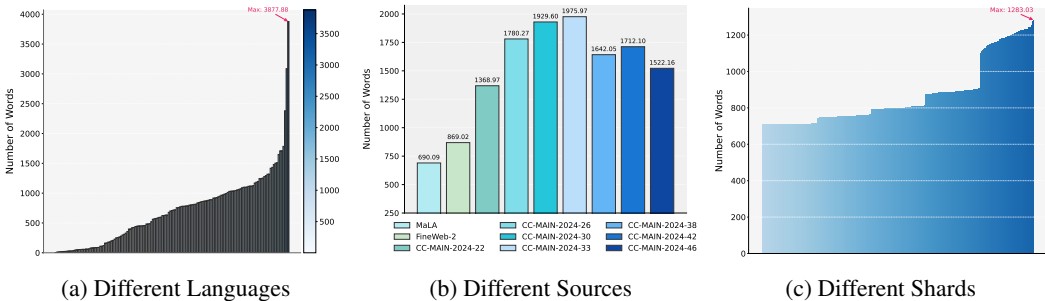

| (a) Different Languages | (b) Different Sources | (c) Different Shards |

Figure 5: Distribution of average word counts across different languages, sources, and shards in the New CC dataset.

Figure 5a illustrates the average word count distribution across different languages in the New CC dataset (CC-MAIN-2024-38). We observe substantial variation in the average word count across languages within the same dataset. For instance, some languages exhibit an average word count as high as 4,000, indicating that their texts are generally longer, while others have an average word count ranging from 50 to 100, suggesting that their texts are typically shorter. This imbalanced distribution complicates the application of traditional fixed-threshold data cleaning methods across all languages. For example, setting a word count threshold of 800 (e.g., the median word count) may be suitable for many languages, but it would still misclassify a significant portion of data as low-quality.

Figure 5b illustrates the average word count distribution for Chinese across different data sources (MaLA, Fineweb-2, and New CC). We observe significant variation in the word count distribution for the same language across these sources. For example, the average word count for Chinese in the MaLA corpus is 690, while in New CC (CC-MAIN-2024-33), the average word count increases to 1,975. This discrepancy highlights the inadequacy of a single fixed threshold for data from different sources. Applying a uniform threshold could lead to incorrect cleaning of Chinese text from certain data sources, potentially compromising the representativeness and quality of the data. Consequently, it is essential to adopt flexible cleaning strategies tailored to the characteristics of each data source.

Figure 5c illustrates the variation in word count for Chinese across different shards in the Fineweb-2 dataset. We observe imbalanced word count distributions between shards, which further complicates the data cleaning process. For instance, some shards contain texts with word counts concentrated between 700 and 1,000, while others have texts primarily between 1,000 and 1,200. This shard-level variation suggests that fixed-threshold cleaning methods may perform inconsistently across different shards, fails to account for the unique characteristics of the data within each shard. Therefore, in the presence of such imbalanced distributions, it is crucial to implement a more flexible data cleaning approach.

# B  DCAD-2000 Grouped by Writting Scripts

As mentioned in Section 4, DCAD-2000 contains a total of 159 writing scripts. To provide a comprehensive overview, we list each of these scripts and their corresponding statistical information in Table 7 and Table 8. By presenting this information, we aim to highlight the broad range of writing systems represented by DCAD and emphasize its potential in various linguistic research and applications.

# C  Data Cleaning Statistics

In this section, we provide detailed data cleaning statistics (Table 9, 10, 11 and 12) for high-resource, medium-resource, and low-resource languages. For the data cleaning statistics of very low-resource languages, please refer to the open-source data statistics we released.

# D    Experimental Setup

In this section, we provide a comprehensive description of our experimental setup, including dataset preparation, model configurations, continued pretraining procedure, data cleaning and anomaly detection pipeline, evaluation metrics, and implementation details.

## D.1    Evaluation Benchmarks

**FineTask Benchmark.**    FineTask is a multilingual, multi-task benchmark covering nine typologically diverse languages: Chinese, French, Arabic, Russian, Thai, Hindi, Turkish, Swahili, and Telugu. The benchmark spans a wide range of NLP tasks including reading comprehension, common-sense reasoning, natural language understanding, and text generation. FineTask provides four evaluation metrics: *Accuracy*, *Accuracy normalized over character length*, *Accuracy normalized over token length*, and *PMI Accuracy*. However, according to statistical data, none of these metrics consistently perform well across all languages. Therefore, we chose to use normalized accuracy (norm accuracy) in our evaluation process.

**Multilingual Benchmarks.**    To analyze performance across varying resource levels, we further evaluate on three established multilingual corpora:

- **SIB-200** [24]: A suite of topic classification datasets across 205 languages. We use the raw accuracy on held-out test sets.
- **Glot500-c** [8]: A curated corpus spanning 500 languages for generation and language modeling. We compute *negative log-likelihood* (NLL) on held-out sentences:

$$\text{NLL} = -\frac{1}{T}\sum_{t=1}^{T}\log p_\theta(w_t \mid w_{<t}), \tag{9}$$

  where $T$ is the total token count in the evaluation set.
- **FLORES-200** [25]: A benchmark for low-resource machine translation covering 200 languages. We translate from English into each target language (Eng-XX) and translate from other languages into English (X-Eng) and evaluate using SacreBLEU with default settings.

## D.2    Pre-training and Evaluation Protocol

We perform continued pretraining on three representative decoder-only large language models (LLMs): LLaMA-3.2-1B [47], Qwen-2.5-7B [48], and Aya-expanse-32B [49]. These models are selected to represent a diverse range of open-source models across different parameter scales, allowing us to investigate the effects of the different dataset, data cleaning pipeline across small, medium, and large model sizes. All models are accessed and managed through the HuggingFace Transformers library.

Given the limitations of computational resources, we refrain from full-parameter finetuning. Instead, we adopt Low-Rank Adaptation (LoRA; 55), a parameter-efficient fine-tuning technique that introduces trainable low-rank matrices into each transformer layer, substantially reducing the number of trainable parameters while maintaining competitive performance.

Our training pipeline closely follows the setup described in the LightEval repository[18], a lightweight evaluation and fine-tuning framework developed by HuggingFace. This ensures reproducibility and consistency with widely adopted community practices. Experiments are conducted on NVIDIA A100 GPU with 80GB memory, which provides sufficient memory bandwidth and compute capability to support batch-level parallelism and efficient LoRA-based fine-tuning. All hyperparameters and task-specific configurations are aligned with those used in the Fineweb-2 benchmark, ensuring comparability with previous work and consistent evaluation conditions.

## D.3    Statistical Anomaly Detection

We provide a detailed comparison of the anomaly detection algorithms evaluated for data cleaning in DCAD-2000. The methods are selected based on their popularity, conceptual diversity, and availability in `scikit-learn`[19]. All experiments are conducted using the same eight-dimensional feature vectors described in Section 3.

**Isolation Forest (ISO_Forest)**    [46] is an ensemble-based method that isolates anomalies instead of profiling normal data points. It constructs random binary trees by recursively selecting features and split values, and then uses the path length of each data point across the trees to assess anomaly scores. Shorter paths indicate

---

[18]`https://github.com/huggingface/lighteval`
[19]`https://scikit-learn.org`

higher likelihood of being an outlier. ISO_Forest is well-suited to high-dimensional and noisy data and requires minimal hyperparameter tuning. Its main drawback is higher computational cost relative to simpler methods, though it scales well with the number of samples.

**One-Class SVM (OC_SVM)**    [50] is a kernel-based method that attempts to separate the data from the origin in a transformed feature space. It is sensitive to the choice of kernel function (e.g., RBF, linear) and associated parameters (e.g., gamma, nu). OC_SVM can be effective in capturing complex boundaries, but it often suffers from scalability issues and requires careful parameter tuning, especially in high-dimensional multilingual settings like DCAD-2000.

**Local Outlier Factor (LOF)**    [51] is a density-based method that identifies anomalies based on local density deviation. It compares the local density of a data point with that of its neighbors. Points that have substantially lower density than their neighbors are considered outliers. The performance of LOF depends heavily on the number of neighbors chosen and tends to degrade in high-dimensional spaces due to the curse of dimensionality. It is also computationally expensive for large datasets.

**K-Means**    [52] is a clustering algorithm typically used for unsupervised partitioning of data. For anomaly detection, it is repurposed by measuring the distance of points from their assigned cluster centroids—points that are far from any centroid can be considered anomalous. K-Means is computationally efficient and easy to implement but lacks sensitivity to local structures and does not inherently model outliers. Its effectiveness depends on a suitable choice of the number of clusters.

## E   Manual Quality Evaluation of Cleaning Pipeline

To validate the effectiveness of our cleaning pipeline and to assess the residual noise and false positives, we conduct a manual quality evaluation of the retained and deleted documents. This evaluation was performed on five representative languages: English, Chinese, German, Japanese, and French. These languages are selected to ensure diverse linguistic coverage, and the evaluation will be extended in future work to include additional languages, particularly low-resource languages, where automatic filtering may be more challenging.

For each of the five languages, we randomly sampled 100 documents retained by our pipeline (i.e., documents that were kept) and 100 documents that were removed (i.e., deleted by our pipeline). The annotation process was conducted by one proficient annotator per language. The key goal of this annotation process was to estimate both the quality of the documents retained by the pipeline and the false positives in the deleted set. The quality evaluation provides insight into how well the cleaning pipeline separates high-quality content from noisy or irrelevant data. The documents were labeled with the following quality ratings:

- **Good:** Documents that were coherent, meaningful, and of high quality.
- **Borderline:** Documents that were understandable but flawed, including minor corruption, weak coherence, or other small issues.
- **Bad:** Documents that were nonsensical, noisy, or semantically meaningless, such as machine translation errors, boilerplate content, spam, or mixed-language noise.

Table 4: **Quality evaluation of retained and deleted documents across five languages.**

| Retained Documents (Kept by filter) | | | | | Deleted Documents (Removed by filter) | | | | |
| --- | --- | --- | --- | --- | --- | --- | --- | --- | --- |
| **Language** | **Good** | **Borderline** | **Bad** | **Residual Noise (Bad %)** | **Language** | **Good** | **Borderline** | **Bad** | **False Positives (Good %)** |
| English | 86% | 10% | 4% | **4%** | English | 5% | 14% | 81% | **5%** |
| Chinese | 82% | 13% | 5% | **5%** | Chinese | 6% | 18% | 76% | **6%** |
| German | 84% | 12% | 4% | **4%** | German | 5% | 16% | 79% | **5%** |
| Japanese | 81% | 12% | 7% | **7%** | Japanese | 6% | 17% | 77% | **6%** |
| French | 84% | 14% | 2% | **2%** | French | 4% | 15% | 81% | **4%** |
| **Avg** | **83.4%** | 12.2% | **4.4%** | **4.4%** | **Avg** | **5.2%** | 16% | **78.8%** | **5.2%** |

Table 4 demonstrate that our cleaning pipeline effectively filters out low-quality content while preserving high-value data. Across all five languages, the proportion of retained documents rated as "Bad" (residual noise) averaged only 4.4%, indicating minimal contamination of retained documents by low-quality content. Similarly, the false positive rate (i.e., representing the proportion of high-quality documents mistakenly removed) was low, averaging 5.2%. The pipeline's precision, defined as the proportion of retained documents classified as "Good" or "Borderline", was 95.6%, while its recall, which measures the retention of "Good" documents, was 94.13%. These results demonstrate that the pipeline achieves both high precision and recall, effectively balancing the removal of noise with the preservation of valuable data. Overall, the findings validate the robustness of our unsupervised, anomaly-detection-based approach across multiple languages, with future work aimed at extending this evaluation to additional languages, particularly low-resource ones.

Table 5: Cost-Benefit Comparison of Filtering Methods (Per 1M Documents)

| Metric | Heuristic Filtering | Anomaly Detection (DCAD) |
|---|---|---|
| Cleaning Time | 10 minutes | 12–15 minutes |
| Max Memory Usage (CPU) | 58 GB | 64 GB |
| Training Data Retained (%) | 88% | 77% |
| Avg Model Accuracy (Global MMLU subset) – LLaMA-3.2-1B | 43.9% | **48.6%** |
| Accuracy Gain per CPU-hour | +0.42% | **+0.60%** |
| Accuracy Gain per 1% data lost | +0.16% | **+0.32%** |

## F  Cost Benefit Analysis of Cleaning Strategies

To better evaluate the practical trade-offs between conventional heuristic filtering and our anomaly-based framework (DCAD), we conduct a controlled cost–benefit analysis on one million web documents under identical hardware conditions. As summarized in Table 5, the DCAD pipeline incurs only a minor computational overhead relative to the heuristic baseline (i.e., approximately two additional minutes of processing time and a 6 GB increase in peak memory usage). Although DCAD retains 11% fewer documents, it consistently yields superior downstream task performance (Section 5), highlighting its effectiveness in balancing data quality and computational efficiency.

Table 6: Ablation Study: Impact of Feature Subsets (Refer to Section 3.2.1)

| Feature Subset Used | Arabic | Turkish |
|---|---|---|
| All 8 features (1–8) | **0.21** | **0.27** |
| w/o (8) Perplexity | 0.20 | 0.25 |
| w/o (7) LID score | **0.16** | **0.21** |
| w/o (6) Flagged word ratio | 0.20 | 0.24 |
| w/o (5) Stopword ratio | 0.21 | 0.26 |
| w/o (4) Special character ratio | 0.20 | 0.26 |
| w/o (3) Word repetition | 0.18 | 0.24 |
| w/o (2) Character repetition | 0.19 | 0.25 |
| w/o (1) Token count | 0.20 | 0.24 |

## G  Feature Robustness Analysis

To evaluate the robustness of our anomaly detection framework with respect to feature design (Section 3.2.1), we conduct a one-feature-at-a-time ablation study. Given the combinatorial explosion of all possible subsets ($2^8 - 1 = 255$), we adopt a pragmatic protocol in which each feature is removed individually from the full 8-dimensional feature vector, and the cleaning process is repeated using the remaining seven features. We then fine-tune LLaMA-3.2-1B on each resulting filtered corpus and evaluate performance on FineTask–Arabic and FineTask–Turkish, following the same experimental setup as in Section 5.

As observed in Table 6, we have the following findings: (1) The full 8-feature configuration consistently outperforms all ablated variants, confirming that each feature contributes meaningfully to overall performance. (2) The Language Identification (LID) confidence score (Feature 7) is particularly critical: its removal results in a substantial accuracy drop, likely due to the presence of mixed or misidentified language content that adversely affects multilingual model quality. (3) Other features, such as repetition ratios and perplexity, provide modest gains individually; none are harmful or redundant when considered in isolation.

## H  Practical Choice of Anomaly Detector and Future Extensions

While modern deep anomaly detection methods, such as autoencoder-based reconstruction scoring [56] and contrastive outlier detection [57], have achieved strong performance in other domains, we deliberately adopt a classical algorithm, specifically Isolation Forest, in this work. This choice is motivated by three practical constraints inherent to large-scale multilingual corpus cleaning:

- **Lack of a clean reference distribution.** Autoencoder-based methods assume access to a predominantly clean training set to learn a reliable reconstruction prior. In our weakly supervised scenario covering 2,282 languages without dependable clean subsets, this assumption is violated, making such models prone to degenerate reconstructions on noisy data.

- **Scalability across languages without supervision.** Contrastive-learning-based outlier detection requires either labeled normal/abnormal pairs or implicitly curated positive anchors. Providing such supervision for thousands of languages would reintroduce the language-specific manual tuning that our language-agnostic pipeline explicitly avoids.

- **Resource efficiency and feature interpretability.** Our framework relies on explicit, interpretable quality features (e.g., repetition ratio, perplexity, LID confidence) rather than opaque embedding-space distances. Classical anomaly detectors like Isolation Forest can operate directly on these CPU-computable features and scale to 46 TB of multilingual data without GPU dependency, making them well-suited for real-world data curation pipelines.

Nonetheless, extending DCAD to incorporate semantic embedding-based anomaly signals or lightweight deep novelty detection represents a promising direction for future work. We view our current feature-space approach as a foundational layer, onto which richer semantic detectors can be incrementally integrated once computational and language-coverage challenges are addressed.

# I   Ethics Statement

Our dataset integrates existing multilingual datasets, such as MaLA [11] and Fineweb-2 [14], and includes newly extracted data from Common Crawl, providing large-scale and high-quality training corpora to support the training of multilingual large language models (LLMs). Additionally, we propose a novel data cleaning method to filter out potentially toxic documents, reducing potential ethical concerns. However, performing fine-grained analysis on such a vast dataset (46.72TB) remains a significant challenge. To address this, we released the dataset for the community to explore and research extensively. Furthermore, since our dataset is derived from open-source datasets, we adhere to the open-source policies of these datasets to promote future research in multilingual LLMs, while mitigating potential ethical risks. Therefore, we believe our dataset does not pose greater societal risks than existing multilingual datasets.

Table 7: **Statisticals grouped by writing scripts (part I).** Comparison of language count, document count, token count, disk size, and sources before and after data cleaning in DCAD-2000.

| Script | #Langs | Documents | | | Tokens | | | Disk Size | | | Source |
|---|---|---|---|---|---|---|---|---|---|---|---|
| | | keep | remove | total | keep | remove | total | keep | remove | total | |
| Latn | 1830 | 5.50B | 439.42M | 5.93B | 4.29T | 327.39B | 4.61T | 21.13TB | 4.91TB | 26.12TB | Fineweb-2, Fineweb, MaLA, New CC |
| Cyrl | 91 | 1.11B | 85.84M | 1.19B | 1.26T | 98.88B | 1.36T | 9.40TB | 2.43TB | 11.83TB | Fineweb-2, MaLA, New CC |
| Hani | 12 | 715.15M | 71.29M | 786.45M | 746.48B | 73.89B | 820.36B | 2.90TB | 1.60TB | 4.50TB | Fineweb-2, MaLA, New CC |
| Jpan | 1 | 491.47M | 42.47M | 533.93M | 278.14B | 22.81B | 300.95B | 2.00TB | 504.87GB | 2.50TB | Fineweb-2, New CC |
| Arab | 60 | 198.64M | 20.36M | 219.03M | 122.36B | 13.12B | 135.48B | 1.03TB | 290.11GB | 1.31TB | Fineweb-2, MaLA, New CC |
| Hang | 1 | 79.22M | 6.16M | 85.38M | 59.07B | 4.62B | 63.69B | 336.56GB | 66.70GB | 403.26GB | Fineweb-2, MaLA, New CC |
| Grek | 4 | 69.14M | 5.90M | 75.04M | 58.45B | 5.15B | 63.60B | 432.64GB | 120.10GB | 552.76GB | Fineweb-2, MaLA, New CC |
| Deva | 48 | 60.09M | 5.79M | 65.87M | 30.63B | 2.56B | 33.19B | 342.83GB | 72.51GB | 415.37GB | Fineweb-2, MaLA, New CC |
| Thai | 11 | 55.73M | 4.34M | 60.06M | 46.36B | 3.60B | 49.96B | 526.40GB | 110.69GB | 637.11GB | Fineweb-2, MaLA, New CC |
| Mlym | 6 | 39.16M | 3.89M | 43.05M | 7.00B | 559.61M | 7.56B | 94.53GB | 18.86GB | 113.40GB | Fineweb-2, MaLA, New CC |
| Gujr | 2 | 38.91M | 4.55M | 43.46M | 5.07B | 461.70M | 5.53B | 60.22GB | 13.54GB | 73.76GB | Fineweb-2, MaLA, New CC |
| Knda | 2 | 34.20M | 2.70M | 36.90M | 4.76B | 359.45M | 5.12B | 68.85GB | 11.14GB | 79.99GB | Fineweb-2, MaLA, New CC |
| Hebr | 6 | 26.99M | 1.83M | 28.82M | 21.15B | 1.38B | 22.53B | 152.34GB | 30.80GB | 183.18GB | Fineweb-2, MaLA, New CC |
| Taml | 2 | 26.65M | 2.92M | 29.56M | 5.88B | 461.60M | 6.35B | 80.38GB | 19.44GB | 99.82GB | Fineweb-2, MaLA, New CC |
| Guru | 2 | 24.04M | 3.16M | 27.21M | 2.27B | 227.69M | 2.50B | 26.71GB | 8.65GB | 35.36GB | Fineweb-2, MaLA, New CC |
| Beng | 6 | 21.91M | 1.51M | 23.42M | 12.67B | 875.46M | 13.54B | 148.42GB | 31.39GB | 179.83GB | Fineweb-2, MaLA, New CC |
| Geor | 3 | 20.56M | 1.36M | 21.92M | 6.19B | 419.61M | 6.61B | 83.04GB | 15.75GB | 98.81GB | Fineweb-2, MaLA, New CC |
| Armn | 4 | 17.24M | 1.46M | 18.70M | 4.74B | 407.38M | 5.15B | 42.47GB | 11.43GB | 53.93GB | Fineweb-2, MaLA, New CC |
| Telu | 4 | 9.93M | 821.21K | 10.75M | 3.91B | 295.72M | 4.20B | 48.22GB | 9.65GB | 57.87GB | Fineweb-2, MaLA, New CC |
| Sinh | 1 | 9.91M | 1.12M | 11.03M | 2.93B | 251.40M | 3.18B | 32.73GB | 7.64GB | 40.37GB | Fineweb-2, MaLA, New CC |
| Orya | 6 | 6.57M | 616.98K | 7.18M | 464.57M | 37.89M | 502.46M | 9.79GB | 2.20GB | 12.01GB | Fineweb-2, MaLA |
| Ethi | 13 | 6.41M | 429.99K | 6.85M | 1.38B | 91.75M | 1.46B | 12.66GB | 2.92GB | 15.59GB | Fineweb-2, MaLA, New CC |
| Mymr | 9 | 6.04M | 479.44K | 6.52M | 5.30B | 406.67M | 5.72B | 40.57GB | 7.83GB | 48.39GB | Fineweb-2, MaLA, New CC |
| Kana | 1 | 5.83M | 1.11M | 6.94M | 1.13B | 219.26M | 1.35B | 16.90GB | 14.33GB | 31.23GB | Fineweb-2, New CC |
| Khmr | 7 | 4.96M | 380.38K | 5.34M | 2.24B | 160.29M | 2.40B | 30.95GB | 4.99GB | 35.95GB | Fineweb-2, MaLA, New CC |
| Bamu | 1 | 4.71M | 1.00M | 5.71M | 199.46M | 42.49M | 241.95M | 79.67GB | 19.47GB | 99.14GB | Fineweb-2, New CC |
| Copt | 2 | 4.40M | 361.99K | 4.76M | 219.04M | 18.03M | 237.09M | 8.97GB | 864.17MB | 9.84GB | Fineweb-2, New CC |
| Tang | 1 | 3.94M | 741.81K | 4.68M | 209.68M | 39.47M | 249.15M | 22.70GB | 7.67GB | 30.36GB | Fineweb-2, New CC |
| Xsux | 1 | 3.90M | 694.59K | 4.59M | 276.93M | 49.35M | 326.28M | 13.84GB | 9.74GB | 23.58GB | Fineweb-2, New CC |
| Laoo | 5 | 3.46M | 470.52K | 3.92M | 840.28M | 87.36M | 927.65M | 11.85GB | 3.95GB | 15.80GB | Fineweb-2, MaLA, New CC |
| Yiii | 1 | 3.39M | 417.38K | 3.81M | 232.88M | 28.68M | 261.56M | 25.82GB | 6.24GB | 32.05GB | Fineweb-2, New CC |
| Hira | 1 | 2.78M | 579.38K | 3.36M | 361.77M | 75.28M | 437.05M | 4.87GB | 4.04GB | 8.91GB | Fineweb-2, New CC |
| Thaa | 2 | 2.51M | 301.28K | 2.82M | 425.90M | 45.08M | 470.98M | 4.75GB | 1.28GB | 6.02GB | Fineweb-2, MaLA, New CC |
| Kits | 1 | 1.86M | 315.45K | 2.17M | 269.54M | 45.75M | 315.29M | 12.47GB | 17.12GB | 29.58GB | Fineweb-2, New CC |
| Hluw | 1 | 1.71M | 374.92K | 2.09M | 70.77M | 15.47M | 86.25M | 3.19GB | 3.45GB | 6.64GB | Fineweb-2, New CC |
| Japn | 1 | 1.60M | 177.40K | 1.78M | 148.77M | 17.99M | 166.76M | 6.05GB | 2.16GB | 8.21GB | MaLA |
| Shrd | 1 | 1.41M | 216.59K | 1.62M | 130.80M | 20.13M | 150.93M | 6.06GB | 2.35GB | 8.40GB | Fineweb-2, New CC |
| Lina | 1 | 1.37M | 271.63K | 1.64M | 130.39M | 25.87M | 156.26M | 6.97GB | 3.85GB | 10.82GB | Fineweb-2, New CC |
| Samr | 1 | 1.35M | 158.99K | 1.51M | 64.06M | 7.54M | 71.59M | 4.30GB | 1.72GB | 6.02GB | Fineweb-2, New CC |
| Cans | 12 | 1.24M | 248.84K | 1.49M | 109.29M | 21.66M | 130.96M | 3.55GB | 2.78GB | 6.33GB | Fineweb-2, MaLA |
| Syrc | 4 | 1.12M | 116.18K | 1.23M | 44.70M | 4.75M | 49.44M | 20.70GB | 4.35GB | 25.04GB | Fineweb-2, MaLA |
| Adlm | 1 | 1.12M | 194.29K | 1.32M | 43.63M | 7.55M | 51.18M | 1.10GB | 853.95MB | 1.95GB | Fineweb-2, New CC |
| Egyp | 1 | 1.12M | 190.50K | 1.31M | 97.41M | 16.58M | 113.99M | 2.54GB | 3.52GB | 6.05GB | Fineweb-2, New CC |
| Mend | 1 | 1.03M | 293.72K | 1.32M | 16.58M | 4.75M | 21.33M | 893.39MB | 2.06GB | 2.95GB | Fineweb-2, New CC |
| Linb | 1 | 735.07K | 107.67K | 842.75K | 52.97M | 7.76M | 60.73M | 6.30GB | 997.90MB | 7.30GB | Fineweb-2, New CC |
| Brai | 1 | 590.10K | 125.33K | 715.43K | 57.85M | 12.29M | 70.13M | 1.94GB | 1.30GB | 3.24GB | Fineweb-2, New CC |
| Sgnw | 1 | 567.29K | 106.45K | 673.74K | 37.34M | 7.01M | 44.34M | 1.40GB | 1.11GB | 2.50GB | Fineweb-2, New CC |
| Tibt | 4 | 544.99K | 70.33K | 615.32K | 288.24M | 33.57M | 321.81M | 4.50GB | 1.53GB | 6.09GB | Fineweb-2, MaLA, New CC |
| Hung | 1 | 520.10K | 155.23K | 675.33K | 42.34M | 12.64M | 54.98M | 1.94GB | 2.32GB | 4.25GB | Fineweb-2, New CC |
| Mong | 3 | 435.35K | 61.47K | 496.83K | 119.66M | 16.95M | 136.62M | 1.97GB | 1.04GB | 3.03GB | Fineweb-2, MaLA |
| Bali | 1 | 422.49K | 77.08K | 499.57K | 39.62M | 7.23M | 46.84M | 1.19GB | 662.91MB | 1.85GB | Fineweb-2, New CC |
| Nshu | 1 | 419.71K | 89.40K | 509.11K | 38.53M | 8.21M | 46.74M | 993.06MB | 1.28GB | 2.27GB | Fineweb-2, New CC |
| Modi | 1 | 386.82K | 67.33K | 454.15K | 52.58M | 9.15M | 61.73M | 16.45GB | 7.42GB | 23.87GB | Fineweb-2, New CC |
| Lana | 1 | 377.58K | 110.80K | 488.38K | 47.55M | 13.95M | 61.50M | 688.16MB | 2.05GB | 2.74GB | Fineweb-2, New CC |
| Saur | 1 | 315.78K | 73.82K | 389.60K | 15.26M | 3.57M | 18.83M | 398.55MB | 489.07MB | 887.62MB | Fineweb-2, New CC |
| Dupl | 1 | 258.90K | 53.06K | 311.96K | 14.14M | 2.90M | 17.04M | 752.58MB | 502.95MB | 1.26GB | Fineweb-2, New CC |
| Runr | 2 | 252.18K | 39.00K | 291.19K | 154.68M | 23.92M | 178.61M | 1.25GB | 3.28GB | 4.52GB | Fineweb-2, MaLA |
| Vaii | 1 | 243.47K | 93.27K | 336.73K | 71.28M | 27.31M | 98.59M | 513.30MB | 1.88GB | 2.39GB | Fineweb-2, New CC |
| Glag | 1 | 237.68K | 72.07K | 309.75K | 20.38M | 6.18M | 26.56M | 476.61MB | 951.96MB | 1.43GB | Fineweb-2, New CC |
| Dsrt | 1 | 198.00K | 37.90K | 235.90K | 4.47M | 855.49K | 5.32M | 248.83MB | 562.92MB | 811.75MB | Fineweb-2, New CC |
| Mroo | 1 | 186.14K | 22.85K | 208.99K | 6.42M | 788.69K | 7.21M | 2.43GB | 335.38MB | 2.77GB | Fineweb-2, New CC |
| Bopo | 1 | 181.71K | 24.45K | 206.16K | 30.63M | 4.12M | 34.75M | 3.45GB | 890.68MB | 4.35GB | Fineweb-2, New CC |
| Mtei | 2 | 175.69K | 20.34K | 196.03K | 49.11M | 5.76M | 54.87M | 805.36MB | 574.03MB | 1.38GB | Fineweb-2, MaLA |
| Khar | 1 | 153.37K | 40.04K | 193.41K | 6.75M | 1.76M | 8.52M | 250.30MB | 182.38MB | 432.67MB | Fineweb-2, New CC |
| Brah | 1 | 138.03K | 22.72K | 160.75K | 7.85M | 1.29M | 9.15M | 273.71MB | 243.75MB | 517.47MB | Fineweb-2, New CC |
| Bhks | 1 | 131.90K | 27.03K | 158.93K | 3.93M | 805.58K | 4.74M | 190.96MB | 154.63MB | 345.59MB | Fineweb-2, New CC |
| Hmnp | 1 | 118.87K | 12.33K | 131.20K | 6.83M | 708.37K | 7.54M | 436.28MB | 151.81MB | 588.09MB | Fineweb-2, New CC |
| Phag | 1 | 107.75K | 17.58K | 125.34K | 3.41M | 556.36K | 3.97M | 141.68MB | 93.31MB | 234.99MB | Fineweb-2, New CC |
| Merc | 1 | 107.52K | 38.04K | 145.56K | 7.61M | 2.69M | 10.30M | 215.43MB | 472.23MB | 687.66MB | Fineweb-2, New CC |
| Kali | 2 | 105.87K | 24.33K | 130.20K | 1.39M | 319.46K | 1.71M | 105.24MB | 91.45MB | 196.70MB | Fineweb-2, New CC |
| Plrd | 1 | 104.31K | 21.07K | 125.38K | 5.47M | 1.10M | 6.57M | 214.53MB | 225.25MB | 439.77MB | Fineweb-2, New CC |
| Lisu | 2 | 101.48K | 20.06K | 121.53K | 24.00M | 4.74M | 28.74M | 204.24MB | 527.21MB | 731.45MB | Fineweb-2, New CC |
| Hmng | 1 | 101.02K | 23.34K | 124.36K | 5.37M | 1.24M | 6.61M | 153.20MB | 196.99MB | 350.19MB | Fineweb-2, New CC |
| Nkoo | 2 | 98.77K | 25.89K | 124.65K | 4.91M | 1.07M | 5.98M | 2.13GB | 233.87MB | 2.36GB | Fineweb-2, MaLA |
| Gran | 1 | 97.96K | 21.57K | 119.53K | 3.57M | 785.93K | 4.36M | 135.27MB | 243.90MB | 379.18MB | Fineweb-2, New CC |
| Gonm | 1 | 94.82K | 16.28K | 111.10K | 2.83M | 486.36K | 3.32M | 106.89MB | 142.16MB | 249.05MB | Fineweb-2, New CC |
| Cher | 2 | 94.19K | 25.99K | 120.19K | 9.12M | 2.45M | 11.57M | 245.29MB | 689.18MB | 934.47MB | Fineweb-2, MaLA |
| Tnsa | 1 | 89.55K | 17.93K | 107.48K | 3.28M | 656.33K | 3.93M | 98.49MB | 204.04MB | 302.53MB | Fineweb-2, New CC |

Table 8: **Statisticals grouped by writing scripts (part II).** Comparison of language count, document count, token count, disk size, and sources before and after data cleaning in DCAD-2000.

| Script | #Langs | Documents | | | Tokens | | | Disk Size | | | Source |
|---|---|---|---|---|---|---|---|---|---|---|---|
| | | keep | remove | total | keep | remove | total | keep | remove | total | |
| Cprt | 1 | 88.19K | 14.11K | 102.30K | 7.87M | 1.26M | 9.13M | 142.36MB | 85.91MB | 228.27MB | Fineweb-2, New CC |
| Cari | 1 | 77.73K | 18.09K | 95.82K | 1.73M | 401.78K | 2.13M | 89.37MB | 76.01MB | 165.38MB | Fineweb-2, New CC |
| Diak | 1 | 68.42K | 22.40K | 90.82K | 2.87M | 938.52K | 3.81M | 58.40MB | 94.36MB | 152.76MB | Fineweb-2, New CC |
| Marc | 1 | 67.80K | 11.89K | 79.69K | 2.34M | 410.50K | 2.75M | 66.51MB | 95.34MB | 161.85MB | Fineweb-2, New CC |
| Mani | 1 | 65.94K | 9.56K | 75.50K | 6.27M | 908.84K | 7.17M | 128.39MB | 140.35MB | 268.75MB | Fineweb-2, New CC |
| Talu | 2 | 65.77K | 11.95K | 77.72K | 1.27M | 231.55K | 1.50M | 78.51MB | 62.21MB | 140.72MB | Fineweb-2, MaLA |
| Vith | 1 | 65.14K | 12.13K | 77.28K | 2.49M | 464.49K | 2.96M | 124.41MB | 95.26MB | 219.66MB | Fineweb-2, New CC |
| Nagm | 1 | 63.57K | 11.94K | 75.51K | 1.03M | 193.45K | 1.22M | 58.20MB | 73.87MB | 132.08MB | Fineweb-2, New CC |
| Ahom | 1 | 60.21K | 9.69K | 69.90K | 2.34M | 376.34K | 2.72M | 127.53MB | 70.68MB | 198.21MB | Fineweb-2, New CC |
| Java | 1 | 58.52K | 13.32K | 71.84K | 2.18M | 496.30K | 2.68M | 66.55MB | 116.13MB | 182.68MB | Fineweb-2, New CC |
| Palm | 1 | 48.99K | 5.32K | 54.32K | 424.13K | 46.09K | 470.22K | 39.41MB | 43.82MB | 83.23MB | Fineweb-2, New CC |
| Wara | 1 | 46.80K | 9.12K | 55.92K | 1.47M | 286.76K | 1.76M | 58.48MB | 52.76MB | 111.24MB | Fineweb-2, New CC |
| Olck | 2 | 45.80K | 4.06K | 49.86K | 6.69M | 492.54K | 7.19M | 86.16MB | 38.55MB | 124.71MB | Fineweb-2, MaLA |
| Khoj | 1 | 39.85K | 5.23K | 45.09K | 892.46K | 117.20K | 1.01M | 43.07MB | 40.20MB | 83.27MB | Fineweb-2, New CC |
| Rohg | 1 | 35.21K | 5.32K | 40.53K | 534.34K | 80.72K | 615.05K | 36.76MB | 41.06MB | 77.82MB | Fineweb-2, New CC |
| Sidd | 1 | 34.75K | 8.41K | 43.16K | 3.03M | 732.80K | 3.76M | 46.06MB | 93.44MB | 139.51MB | Fineweb-2, New CC |
| Yezi | 1 | 33.92K | 3.35K | 37.27K | 96.61K | 9.53K | 106.13K | 29.36MB | 14.31MB | 43.67MB | Fineweb-2, New CC |
| Ougr | 1 | 32.34K | 6.13K | 38.47K | 442.16K | 83.82K | 525.98K | 31.03MB | 37.95MB | 68.98MB | Fineweb-2, New CC |
| Avst | 1 | 32.16K | 6.62K | 38.78K | 1.75M | 360.09K | 2.11M | 51.64MB | 53.81MB | 105.46MB | Fineweb-2, New CC |
| Ital | 1 | 32.06K | 5.06K | 37.12K | 519.27K | 81.93K | 601.19K | 34.30MB | 29.24MB | 63.53MB | Fineweb-2, New CC |
| Wcho | 1 | 31.94K | 6.51K | 38.45K | 1.48M | 301.04K | 1.78M | 58.25MB | 74.54MB | 132.79MB | Fineweb-2, New CC |
| Kthi | 1 | 31.07K | 5.44K | 36.51K | 763.52K | 133.75K | 897.27K | 30.79MB | 35.73MB | 66.52MB | Fineweb-2, New CC |
| Tavt | 1 | 30.95K | 3.63K | 34.57K | 670.82K | 78.65K | 749.47K | 29.30MB | 14.97MB | 44.26MB | Fineweb-2, New CC |
| Takr | 1 | 30.70K | 5.29K | 35.99K | 1.73M | 298.02K | 2.03M | 30.89MB | 45.59MB | 76.48MB | Fineweb-2, New CC |
| Tfng | 4 | 29.84K | 3.34K | 33.18K | 1.42M | 148.55K | 1.57M | 35.12MB | 24.87MB | 59.99MB | Fineweb-2, New CC |
| Tale | 1 | 26.17K | 2.80K | 28.98K | 220.84K | 23.64K | 244.48K | 23.80MB | 16.84MB | 40.64MB | Fineweb-2, New CC |
| Elba | 1 | 24.86K | 4.61K | 29.48K | 394.51K | 73.22K | 467.73K | 24.19MB | 19.19MB | 43.38MB | Fineweb-2, New CC |
| Zanb | 1 | 24.46K | 4.76K | 29.21K | 327.39K | 63.68K | 391.07K | 26.07MB | 40.03MB | 66.10MB | Fineweb-2, New CC |
| Sogo | 1 | 22.29K | 3.88K | 26.16K | 146.13K | 25.41K | 171.54K | 17.82MB | 20.07MB | 37.89MB | Fineweb-2, New CC |
| Soyo | 1 | 22.21K | 4.91K | 27.12K | 598.89K | 132.47K | 731.36K | 25.04MB | 36.77MB | 61.81MB | Fineweb-2, New CC |
| Dogr | 1 | 21.29K | 3.82K | 25.11K | 1.28M | 229.94K | 1.51M | 29.94MB | 23.89MB | 53.84MB | Fineweb-2, New CC |
| Kawi | 1 | 20.28K | 4.10K | 24.38K | 396.57K | 80.26K | 476.83K | 20.90MB | 24.30MB | 45.20MB | Fineweb-2, New CC |
| Phli | 1 | 19.16K | 2.88K | 22.04K | 41.16K | 6.19K | 47.35K | 17.52MB | 7.60MB | 25.13MB | Fineweb-2, New CC |
| Cham | 1 | 17.92K | 3.60K | 21.52K | 762.24K | 153.32K | 915.57K | 21.12MB | 39.91MB | 61.03MB | Fineweb-2, New CC |
| Nbat | 1 | 17.61K | 3.19K | 20.80K | 280.13K | 50.76K | 330.89K | 18.90MB | 15.97MB | 34.87MB | Fineweb-2, New CC |
| Nand | 1 | 17.39K | 3.36K | 20.75K | 307.12K | 59.32K | 366.44K | 17.76MB | 19.20MB | 36.96MB | Fineweb-2, New CC |
| Osma | 1 | 16.98K | 2.59K | 19.57K | 495.54K | 75.61K | 571.15K | 19.16MB | 15.11MB | 34.27MB | Fineweb-2, New CC |
| Sind | 1 | 14.81K | 4.24K | 19.05K | 315.61K | 90.31K | 405.93K | 21.16MB | 18.70MB | 39.86MB | Fineweb-2, New CC |
| Sogd | 1 | 14.52K | 2.73K | 17.24K | 307.50K | 57.79K | 365.30K | 14.67MB | 9.73MB | 24.40MB | Fineweb-2, New CC |
| Pauc | 1 | 13.23K | 4.28K | 17.50K | 1.88M | 609.43K | 2.49M | 13.65MB | 33.03MB | 46.67MB | Fineweb-2, New CC |
| Sylo | 1 | 12.42K | 2.88K | 15.29K | 922.71K | 213.86K | 1.14M | 22.76MB | 22.23MB | 44.99MB | Fineweb-2, New CC |
| Goth | 2 | 11.84K | 1.24K | 13.08K | 191.30K | 19.67K | 210.97K | 11.59MB | 3.62MB | 15.22MB | Fineweb-2, MaLA |
| Rjng | 1 | 10.30K | 2.36K | 12.65K | 595.51K | 136.27K | 731.78K | 9.43MB | 15.02MB | 24.45MB | Fineweb-2, New CC |
| Chrs | 1 | 10.24K | 1.26K | 11.50K | 45.98K | 5.66K | 51.64K | 8.22MB | 5.45MB | 13.67MB | Fineweb-2, New CC |
| Phlp | 1 | 9.08K | 2.03K | 11.11K | 31.62K | 7.06K | 38.69K | 8.35MB | 5.61MB | 13.96MB | Fineweb-2, New CC |
| Mand | 1 | 8.73K | 1.49K | 10.21K | 82.87K | 14.11K | 96.98K | 9.07MB | 5.24MB | 14.31MB | Fineweb-2, New CC |
| Tglg | 1 | 8.58K | 1.88K | 10.46K | 638.75K | 140.15K | 778.89K | 11.22MB | 10.89MB | 22.11MB | Fineweb-2, New CC |
| Shaw | 1 | 8.41K | 1.28K | 9.69K | 915.43K | 139.72K | 1.06M | 13.65MB | 12.62MB | 26.27MB | Fineweb-2, New CC |
| Hatr | 1 | 7.44K | 1.63K | 9.07K | 371.48K | 81.61K | 453.09K | 10.15MB | 13.53MB | 23.68MB | Fineweb-2, New CC |
| Bugi | 2 | 7.03K | 1.33K | 8.36K | 95.81K | 18.11K | 113.91K | 6.90MB | 6.18MB | 13.09MB | Fineweb-2, MaLA |
| Tagb | 1 | 6.58K | 1.14K | 7.72K | 30.92K | 5.37K | 36.30K | 5.84MB | 2.33MB | 8.17MB | Fineweb-2, New CC |
| Prti | 1 | 6.05K | 1.09K | 7.15K | 225.93K | 40.79K | 266.72K | 7.31MB | 4.57MB | 11.89MB | Fineweb-2, New CC |
| Narb | 1 | 5.22K | 835 | 6.06K | 56.09K | 8.97K | 65.06K | 6.01MB | 7.12MB | 13.13MB | Fineweb-2, New CC |
| Sarb | 1 | 4.99K | 874 | 5.86K | 170.46K | 29.86K | 200.31K | 6.93MB | 15.95MB | 22.87MB | Fineweb-2, New CC |
| Ugar | 1 | 4.85K | 653 | 5.50K | 133.05K | 17.92K | 150.97K | 4.03MB | 2.47MB | 6.50MB | Fineweb-2, New CC |
| Lydi | 1 | 4.59K | 1.03K | 5.62K | 28.08M | 6.29M | 34.37M | 77.22MB | 70.99MB | 148.21MB | Fineweb-2, New CC |
| Buhd | 1 | 3.16K | 448 | 3.61K | 7.77K | 1.10K | 8.87K | 2.73MB | 623.88KB | 3.35MB | Fineweb-2, New CC |
| Perm | 1 | 2.87K | 630 | 3.50K | 19.17K | 4.20K | 23.37K | 2.58MB | 1.36MB | 3.94MB | Fineweb-2, New CC |
| Elym | 1 | 1.66K | 496 | 2.16K | 61.25K | 18.28K | 79.53K | 1.88MB | 7.52MB | 9.40MB | Fineweb-2, New CC |
| Limb | 1 | 59 | 15 | 74 | 32.32K | 8.22K | 40.53K | 754.75KB | 229.80KB | 984.54KB | Fineweb-2, New CC |

Table 9: **Data Cleaning Statistics (part I):** Comparison of document count, token count, disk size, and sources before and after data cleaning in DCAD-2000.

| Lang Code | Documents | | | Tokens | | | Disk Size | | | Source |
|---|---|---|---|---|---|---|---|---|---|---|
| | keep | remove | total | keep | remove | total | keep | remove | total | |
| eng_Latn | 1.31B | 101.08M | 1.41B | 1.21T | 93.23B | 1.30T | 5.66TB | 1.49TB | 7.15TB | Fineweb, MaLA, New CC |
| rus_Cyrl | 858.53M | 67.21M | 925.74M | 1.14T | 90.18B | 1.23T | 8.40TB | 2.22TB | 10.62TB | Fineweb-2, MaLA, New CC |
| cmn_Hani | 713.97M | 71.19M | 785.16M | 745.88B | 73.84B | 819.71B | 2.90TB | 1.60TB | 4.50TB | Fineweb-2, New CC |
| deu_Latn | 668.62M | 53.65M | 722.27M | 632.32B | 51.11B | 683.44B | 2.85TB | 664.79GB | 3.52TB | Fineweb-2, MaLA, New CC |
| spa_Latn | 604.45M | 43.33M | 647.79M | 483.75B | 34.79B | 518.54B | 2.54TB | 498.55GB | 3.03TB | Fineweb-2, MaLA, New CC |
| fra_Latn | 513.53M | 40.32M | 553.85M | 430.86B | 33.77B | 464.64B | 2.15TB | 491.23GB | 2.64TB | Fineweb-2, MaLA, New CC |
| jpn_Jpan | 491.47M | 42.47M | 533.93M | 278.14B | 22.81B | 300.95B | 2.00TB | 504.87GB | 2.50TB | Fineweb-2, New CC |
| ita_Latn | 311.42M | 25.50M | 336.93M | 250.12B | 20.69B | 270.81B | 1.29TB | 292.75GB | 1.59TB | Fineweb-2, MaLA, New CC |
| por_Latn | 271.48M | 18.67M | 290.15M | 204.64B | 14.12B | 218.77B | 1.07TB | 225.83GB | 1.30TB | Fineweb-2, MaLA, New CC |
| pol_Latn | 223.21M | 15.38M | 238.59M | 180.34B | 12.59B | 192.93B | 910.55GB | 184.59GB | 1.10TB | Fineweb-2, MaLA, New CC |
| nld_Latn | 219.16M | 14.03M | 233.19M | 146.16B | 9.38B | 155.54B | 739.62GB | 159.01GB | 898.63GB | Fineweb-2, MaLA, New CC |
| ind_Latn | 156.92M | 16.21M | 173.12M | 60.97B | 5.15B | 66.11B | 406.86GB | 64.84GB | 471.70GB | Fineweb-2, MaLA |
| tur_Latn | 143.31M | 9.98M | 153.30M | 118.40B | 8.21B | 126.61B | 618.87GB | 145.39GB | 764.26GB | Fineweb-2, MaLA, New CC |
| vie_Latn | 87.77M | 6.19M | 93.96M | 110.11B | 7.71B | 117.82B | 570.86GB | 116.19GB | 687.05GB | Fineweb-2, MaLA, New CC |
| fas_Arab | 82.80M | 9.49M | 92.29M | 67.58B | 7.91B | 75.49B | 521.39GB | 121.46GB | 642.85GB | Fineweb-2, MaLA, New CC |
| kor_Hang | 79.22M | 6.16M | 85.38M | 59.07B | 4.62B | 63.69B | 336.56GB | 66.70GB | 403.26GB | Fineweb-2, MaLA, New CC |
| swe_Latn | 77.32M | 5.08M | 82.40M | 59.21B | 3.92B | 63.13B | 269.37GB | 73.25GB | 342.62GB | Fineweb-2, MaLA, New CC |
| hun_Latn | 70.79M | 5.18M | 75.97M | 65.62B | 4.86B | 70.48B | 319.58GB | 87.97GB | 407.55GB | Fineweb-2, MaLA, New CC |
| ukr_Cyrl | 67.87M | 4.31M | 72.18M | 53.79B | 3.41B | 57.20B | 428.74GB | 82.51GB | 511.25GB | Fineweb-2, MaLA, New CC |
| ell_Grek | 67.48M | 5.67M | 73.15M | 57.63B | 5.03B | 62.66B | 425.03GB | 112.67GB | 537.71GB | Fineweb-2, MaLA, New CC |
| tha_Thai | 55.47M | 4.29M | 59.76M | 46.31B | 3.59B | 49.90B | 525.54GB | 110.37GB | 635.91GB | Fineweb-2, MaLA, New CC |
| arb_Arab | 53.70M | 4.06M | 57.76M | 25.21B | 1.92B | 27.13B | 278.77GB | 72.25GB | 351.02GB | Fineweb-2, MaLA |
| aze_Latn | 51.38M | 6.44M | 57.82M | 3.30B | 392.00M | 3.70B | 41.90GB | 10.70GB | 52.60GB | MaLA |
| slv_Latn | 50.41M | 4.05M | 54.46M | 11.66B | 836.48M | 12.50B | 69.22GB | 12.64GB | 81.87GB | Fineweb-2, MaLA, New CC |
| cat_Latn | 48.83M | 3.78M | 52.61M | 16.49B | 1.13B | 17.62B | 96.97GB | 14.24GB | 111.21GB | Fineweb-2, MaLA, New CC |
| fin_Latn | 47.80M | 4.09M | 51.89M | 43.43B | 3.75B | 47.19B | 202.14GB | 57.62GB | 259.76GB | Fineweb-2, MaLA, New CC |
| ces_Latn | 47.54M | 3.21M | 50.74M | 42.20B | 2.84B | 45.04B | 195.62GB | 48.74GB | 244.36GB | MaLA, New CC |
| hbs_Latn | 42.98M | 8.05M | 51.04M | 1.53B | 287.34M | 1.82B | 22.41GB | 6.41GB | 28.82GB | MaLA |
| fil_Latn | 40.15M | 6.32M | 46.47M | 3.47B | 477.70M | 3.94B | 31.22GB | 9.20GB | 40.42GB | Fineweb-2, MaLA |
| mal_Mlym | 39.10M | 3.88M | 42.98M | 7.00B | 558.83M | 7.56B | 94.47GB | 18.30GB | 112.78GB | Fineweb-2, MaLA, New CC |
| nob_Latn | 38.88M | 4.33M | 43.21M | 24.13B | 2.81B | 26.94B | 139.85GB | 66.29GB | 206.15GB | Fineweb-2, MaLA |
| guj_Gujr | 38.82M | 4.54M | 43.36M | 5.07B | 461.54M | 5.53B | 60.08GB | 13.49GB | 73.57GB | Fineweb-2, MaLA, New CC |
| bul_Cyrl | 37.11M | 2.56M | 39.67M | 32.29B | 2.23B | 34.51B | 245.84GB | 55.86GB | 301.69GB | Fineweb-2, MaLA, New CC |
| kan_Knda | 34.20M | 2.70M | 36.90M | 4.76B | 359.21M | 5.12B | 68.82GB | 11.13GB | 79.95GB | Fineweb-2, MaLA, New CC |
| hin_Deva | 29.15M | 2.47M | 31.62M | 22.08B | 1.81B | 23.89B | 219.46GB | 46.45GB | 265.91GB | Fineweb-2, MaLA, New CC |
| tam_Taml | 26.55M | 2.90M | 29.45M | 5.88B | 460.75M | 6.34B | 80.26GB | 19.29GB | 99.55GB | Fineweb-2, MaLA, New CC |
| kaz_Cyrl | 25.78M | 1.67M | 27.45M | 6.37B | 432.67M | 6.80B | 64.36GB | 12.99GB | 77.35GB | Fineweb-2, MaLA, New CC |
| heb_Hebr | 25.24M | 1.61M | 26.85M | 20.74B | 1.33B | 22.07B | 147.85GB | 28.75GB | 176.60GB | Fineweb-2, MaLA, New CC |
| ara_Arab | 25.14M | 3.24M | 28.39M | 17.21B | 2.23B | 19.44B | 152.73GB | 71.93GB | 224.66GB | MaLA, New CC |
| srp_Cyrl | 25.13M | 1.75M | 26.88M | 6.91B | 496.07M | 7.41B | 60.34GB | 8.50GB | 68.84GB | Fineweb-2, MaLA, New CC |
| est_Latn | 24.18M | 2.86M | 27.04M | 2.89B | 294.20M | 3.18B | 26.17GB | 8.91GB | 35.08GB | MaLA, New CC |
| sqi_Latn | 24.16M | 3.25M | 27.41M | 2.38B | 237.81M | 2.61B | 21.08GB | 5.03GB | 26.11GB | MaLA, New CC |
| isl_Latn | 24.06M | 2.23M | 26.29M | 6.32B | 561.74M | 6.89B | 34.88GB | 9.09GB | 43.97GB | Fineweb-2, MaLA, New CC |
| pan_Guru | 24.02M | 3.16M | 27.19M | 2.27B | 227.60M | 2.50B | 26.69GB | 8.59GB | 35.28GB | MaLA, New CC |
| mlt_Latn | 23.37M | 2.08M | 25.45M | 3.24B | 322.80M | 3.56B | 16.40GB | 4.96GB | 21.36GB | Fineweb-2, MaLA, New CC |
| mkd_Cyrl | 22.61M | 1.89M | 24.50M | 5.29B | 396.98M | 5.68B | 51.37GB | 7.08GB | 58.45GB | Fineweb-2, MaLA, New CC |
| bos_Latn | 21.62M | 1.71M | 23.33M | 11.01B | 831.59M | 11.84B | 59.71GB | 10.67GB | 70.38GB | Fineweb-2, MaLA, New CC |
| kat_Geor | 20.27M | 1.30M | 21.57M | 6.16B | 413.36M | 6.57B | 82.54GB | 15.10GB | 97.65GB | Fineweb-2, MaLA, New CC |
| lit_Latn | 20.09M | 1.51M | 21.60M | 17.47B | 1.33B | 18.80B | 91.29GB | 18.30GB | 109.59GB | Fineweb-2, MaLA, New CC |
| ben_Beng | 19.90M | 1.37M | 21.28M | 12.26B | 848.75M | 13.11B | 143.64GB | 30.36GB | 174.00GB | Fineweb-2, MaLA, New CC |
| hrv_Latn | 19.83M | 1.54M | 21.37M | 15.02B | 1.19B | 16.21B | 76.53GB | 16.65GB | 93.18GB | Fineweb-2, MaLA, New CC |
| glg_Latn | 19.31M | 1.58M | 20.89M | 4.45B | 372.72M | 4.83B | 28.40GB | 4.50GB | 32.90GB | Fineweb-2, MaLA, New CC |
| ron_Latn | 18.28M | 1.42M | 19.69M | 23.42B | 1.81B | 25.23B | 110.94GB | 20.14GB | 131.08GB | MaLA, New CC |
| ceb_Latn | 18.14M | 1.82M | 19.97M | 1.91B | 184.52M | 2.09B | 14.11GB | 2.06GB | 16.18GB | Fineweb-2, MaLA, New CC |
| hye_Armn | 16.93M | 1.40M | 18.33M | 4.65B | 392.68M | 5.04B | 41.29GB | 10.76GB | 52.05GB | Fineweb-2, MaLA, New CC |
| msa_Latn | 16.90M | 1.51M | 18.40M | 12.27B | 1.05B | 13.32B | 67.19GB | 34.22GB | 101.42GB | MaLA, New CC |
| tgk_Cyrl | 16.60M | 1.04M | 17.64M | 3.46B | 241.47M | 3.70B | 29.00GB | 5.01GB | 34.01GB | Fineweb-2, MaLA, New CC |
| mar_Deva | 15.37M | 1.35M | 16.72M | 4.05B | 287.28M | 4.34B | 52.49GB | 7.16GB | 59.65GB | Fineweb-2, MaLA, New CC |
| bel_Cyrl | 15.22M | 1.06M | 16.29M | 5.30B | 353.85M | 5.65B | 45.23GB | 6.76GB | 51.99GB | Fineweb-2, MaLA, New CC |
| nep_Deva | 13.18M | 1.74M | 14.91M | 3.40B | 354.95M | 3.75B | 57.72GB | 14.16GB | 71.88GB | MaLA, New CC |
| urd_Arab | 12.92M | 1.28M | 14.20M | 5.63B | 463.49M | 6.09B | 43.36GB | 8.33GB | 51.69GB | Fineweb-2, MaLA, New CC |
| slk_Latn | 12.79M | 850.42K | 13.64M | 10.71B | 712.57M | 11.43B | 53.49GB | 10.01GB | 63.50GB | MaLA, New CC |
| mon_Cyrl | 11.46M | 1.37M | 12.83M | 2.05B | 225.17M | 2.27B | 25.55GB | 7.89GB | 33.44GB | MaLA, New CC |
| dan_Latn | 11.33M | 645.36K | 11.98M | 8.91B | 506.75M | 9.42B | 42.48GB | 9.31GB | 51.78GB | MaLA, New CC |
| eus_Latn | 10.88M | 720.92K | 11.60M | 2.86B | 180.73M | 3.04B | 18.54GB | 2.98GB | 21.52GB | Fineweb-2, MaLA, New CC |
| azj_Latn | 10.37M | 764.57K | 11.14M | 6.02B | 427.97M | 6.45B | 54.46GB | 9.98GB | 64.44GB | Fineweb-2, MaLA, New CC |
| swa_Latn | 10.32M | 1.78M | 12.10M | 968.63M | 131.70M | 1.10B | 8.88GB | 2.59GB | 11.47GB | MaLA, New CC |
| als_Latn | 9.94M | 695.21K | 10.64M | 7.84B | 540.49M | 8.38B | 22.16GB | 3.80GB | 25.97GB | Fineweb-2, MaLA |
| sin_Sinh | 9.91M | 1.12M | 11.03M | 2.93B | 251.40M | 3.18B | 32.73GB | 7.64GB | 40.37GB | Fineweb-2, MaLA, New CC |
| lat_Latn | 9.86M | 968.13K | 10.83M | 1.67B | 209.54M | 1.88B | 8.93GB | 3.35GB | 12.27GB | Fineweb-2, MaLA, New CC |
| tel_Telu | 9.81M | 790.37K | 10.60M | 3.90B | 293.32M | 4.19B | 47.82GB | 9.23GB | 57.05GB | Fineweb-2, MaLA, New CC |
| afr_Latn | 9.38M | 858.54K | 10.24M | 3.02B | 252.81M | 3.27B | 16.05GB | 3.08GB | 19.13GB | Fineweb-2, MaLA, New CC |

Table 10: **Data Cleaning Statistics (part II):** Comparison of document count, token count, disk size, and sources before and after data cleaning in DCAD-2000.

| Lang Code | Documents | | | Tokens | | | Disk Size | | | Source |
|---|---|---|---|---|---|---|---|---|---|---|
| | keep | remove | total | keep | remove | total | keep | remove | total | |
| ekk_Latn | 9.24M | 772.47K | 10.01M | 4.79B | 401.83M | 5.19B | 38.34GB | 11.83GB | 50.16GB | Fineweb-2, MaLA |
| zsm_Latn | 8.67M | 795.54K | 9.47M | 4.22B | 365.48M | 4.59B | 31.54GB | 8.93GB | 40.48GB | Fineweb-2, MaLA |
| ltz_Latn | 8.59M | 1.21M | 9.79M | 1.18B | 146.26M | 1.33B | 6.77GB | 1.91GB | 8.68GB | Fineweb-2, MaLA, New CC |
| som_Latn | 7.47M | 716.70K | 8.19M | 2.20B | 193.46M | 2.40B | 10.27GB | 3.34GB | 13.61GB | Fineweb-2, MaLA, New CC |
| kir_Cyrl | 6.47M | 468.94K | 6.94M | 2.31B | 183.29M | 2.49B | 21.00GB | 3.63GB | 24.63GB | Fineweb-2, MaLA, New CC |
| cym_Latn | 6.47M | 515.43K | 6.99M | 2.01B | 141.85M | 2.15B | 10.29GB | 1.99GB | 12.28GB | Fineweb-2, MaLA, New CC |
| nor_Latn | 6.13M | 733.57K | 6.87M | 1.27B | 150.12M | 1.42B | 8.91GB | 2.74GB | 11.65GB | MaLA, New CC |
| uzb_Latn | 6.07M | 715.37K | 6.78M | 929.54M | 98.71M | 1.03B | 8.76GB | 2.73GB | 11.49GB | MaLA, New CC |
| und_Kana | 5.83M | 1.11M | 6.94M | 1.13B | 219.26M | 1.35B | 16.90GB | 14.33GB | 31.23GB | Fineweb-2, New CC |
| mya_Mymr | 5.80M | 449.02K | 6.25M | 5.28B | 404.36M | 5.69B | 40.05GB | 7.53GB | 47.57GB | Fineweb-2, MaLA, New CC |
| epo_Latn | 5.77M | 456.78K | 6.23M | 2.38B | 177.31M | 2.56B | 12.03GB | 2.25GB | 14.27GB | Fineweb-2, MaLA, New CC |
| ary_Arab | 5.67M | 465.36K | 6.14M | 1.38B | 114.17M | 1.50B | 18.12GB | 4.32GB | 22.44GB | Fineweb-2, MaLA |
| lvs_Latn | 5.51M | 382.81K | 5.89M | 2.74B | 185.99M | 2.92B | 21.58GB | 6.85GB | 28.43GB | Fineweb-2, MaLA |
| hau_Latn | 5.48M | 662.28K | 6.15M | 438.94M | 49.38M | 488.32M | 3.22GB | 1.09GB | 4.32GB | MaLA |
| gle_Latn | 5.47M | 428.92K | 5.90M | 1.65B | 134.54M | 1.78B | 9.41GB | 1.55GB | 10.96GB | Fineweb-2, MaLA, New CC |
| nno_Latn | 5.19M | 553.48K | 5.75M | 1.35B | 124.05M | 1.48B | 7.48GB | 1.85GB | 9.33GB | Fineweb-2, MaLA, New CC |
| ory_Orya | 5.13M | 444.55K | 5.57M | 325.74M | 23.33M | 349.07M | 7.34GB | 1.07GB | 8.41GB | Fineweb-2, MaLA |
| amh_Ethi | 4.86M | 302.32K | 5.17M | 1.21B | 77.95M | 1.28B | 10.27GB | 1.56GB | 11.83GB | Fineweb-2, MaLA, New CC |
| khm_Khmr | 4.74M | 344.10K | 5.09M | 2.23B | 158.45M | 2.39B | 30.49GB | 4.58GB | 35.08GB | Fineweb-2, MaLA, New CC |
| tat_Cyrl | 4.72M | 390.38K | 5.11M | 1.29B | 103.35M | 1.39B | 11.66GB | 2.16GB | 13.82GB | Fineweb-2, MaLA, New CC |
| und_Bamu | 4.71M | 1.00M | 5.71M | 199.46M | 42.49M | 241.95M | 79.67GB | 19.47GB | 99.14GB | Fineweb-2, New CC |
| und_Copt | 4.40M | 361.86K | 4.76M | 218.11M | 17.95M | 236.07M | 8.96GB | 860.56MB | 9.82GB | Fineweb-2, New CC |
| arz_Arab | 4.19M | 347.36K | 4.54M | 794.23M | 62.86M | 857.09M | 6.87GB | 1.16GB | 8.03GB | Fineweb-2, MaLA, New CC |
| und_Tang | 3.94M | 741.81K | 4.68M | 209.68M | 39.47M | 249.15M | 22.70GB | 7.67GB | 30.36GB | Fineweb-2, New CC |
| und_Xsux | 3.90M | 694.59K | 4.59M | 276.93M | 49.35M | 326.28M | 13.84GB | 9.74GB | 23.58GB | Fineweb-2, New CC |
| lav_Latn | 3.76M | 347.11K | 4.11M | 2.12B | 196.45M | 2.31B | 13.96GB | 7.36GB | 21.32GB | MaLA, New CC |
| pus_Arab | 3.71M | 493.24K | 4.21M | 905.77M | 106.28M | 1.01B | 7.66GB | 2.36GB | 10.02GB | MaLA, New CC |
| hbs_Cyrl | 3.47M | 463.55K | 3.93M | 131.15M | 17.53M | 148.69M | 2.47GB | 544.73MB | 3.02GB | MaLA, New CC |
| war_Latn | 3.43M | 283.72K | 3.71M | 137.36M | 11.19M | 148.55M | 1.84GB | 161.55MB | 2.00GB | Fineweb-2, MaLA, New CC |
| und_Yiii | 3.39M | 417.38K | 3.81M | 232.88M | 28.68M | 261.56M | 25.82GB | 6.24GB | 32.05GB | Fineweb-2, New CC |
| multi_Latn | 3.11M | 394.01K | 3.50M | 2.39B | 303.45M | 2.70B | 18.42GB | 7.60GB | 26.02GB | New CC |
| mlg_Latn | 2.85M | 437.74K | 3.29M | 288.34M | 41.29M | 329.63M | 2.74GB | 765.89MB | 3.51GB | MaLA, New CC |
| und_Hira | 2.78M | 579.38K | 3.36M | 361.77M | 75.28M | 437.05M | 4.87GB | 4.04GB | 8.91GB | Fineweb-2, New CC |
| uzn_Cyrl | 2.61M | 304.12K | 2.91M | 396.89M | 30.84M | 427.73M | 6.39GB | 1.47GB | 7.86GB | Fineweb-2, MaLA |
| hat_Latn | 2.58M | 226.91K | 2.81M | 464.18M | 41.25M | 505.43M | 2.60GB | 548.40MB | 3.15GB | Fineweb-2, MaLA, New CC |
| zul_Latn | 2.47M | 294.21K | 2.76M | 333.05M | 38.27M | 371.33M | 2.15GB | 642.83MB | 2.79GB | Fineweb-2, MaLA |
| kur_Latn | 2.41M | 327.93K | 2.74M | 482.02M | 51.67M | 533.69M | 3.40GB | 1.04GB | 4.44GB | MaLA |
| div_Thaa | 2.25M | 263.72K | 2.52M | 418.22M | 43.98M | 462.20M | 4.37GB | 1.02GB | 5.38GB | Fineweb-2, MaLA, New CC |
| tgl_Latn | 2.24M | 345.69K | 2.59M | 369.19M | 35.56M | 404.74M | 2.75GB | 669.71MB | 3.42GB | MaLA, New CC |
| uzb_Cyrl | 2.22M | 314.25K | 2.54M | 194.02M | 27.60M | 221.61M | 2.96GB | 1.14GB | 4.10GB | MaLA |
| fry_Latn | 2.14M | 232.49K | 2.38M | 605.32M | 65.90M | 671.22M | 3.10GB | 914.11MB | 4.01GB | Fineweb-2, MaLA, New CC |
| sna_Latn | 2.14M | 181.61K | 2.32M | 295.33M | 24.54M | 319.87M | 1.84GB | 428.76MB | 2.27GB | Fineweb-2, MaLA |
| fao_Latn | 2.09M | 163.66K | 2.26M | 199.43M | 14.19M | 213.61M | 1.69GB | 392.84MB | 2.08GB | Fineweb-2, MaLA |
| und_Laoo | 2.06M | 364.70K | 2.42M | 212.14M | 37.64M | 249.78M | 4.20GB | 2.59GB | 6.79GB | Fineweb-2, New CC |
| sun_Latn | 1.99M | 193.82K | 2.19M | 275.24M | 25.28M | 300.53M | 1.71GB | 543.58MB | 2.25GB | Fineweb-2, MaLA, New CC |
| snd_Arab | 1.91M | 154.84K | 2.06M | 1.12B | 105.00M | 1.22B | 5.27GB | 1.88GB | 7.15GB | Fineweb-2, MaLA, New CC |
| und_Cyrl | 1.86M | 427.20K | 2.29M | 1.32B | 302.88M | 1.62B | 5.09GB | 18.81GB | 23.90GB | Fineweb-2, New CC |
| und_Kits | 1.86M | 315.45K | 2.17M | 269.54M | 45.75M | 315.29M | 12.47GB | 17.12GB | 29.58GB | Fineweb-2, New CC |
| bak_Cyrl | 1.85M | 132.43K | 1.99M | 401.91M | 27.62M | 429.53M | 3.87GB | 733.50MB | 4.60GB | Fineweb-2, MaLA, New CC |
| asm_Beng | 1.82M | 115.52K | 1.93M | 380.78M | 23.67M | 404.45M | 4.50GB | 907.15MB | 5.40GB | Fineweb-2, MaLA, New CC |
| cos_Latn | 1.79M | 274.66K | 2.06M | 228.06M | 35.24M | 263.31M | 1.10GB | 580.00MB | 1.68GB | MaLA |
| ckb_Arab | 1.78M | 177.88K | 1.96M | 841.60M | 76.59M | 918.19M | 6.48GB | 1.52GB | 8.00GB | Fineweb-2, MaLA, New CC |
| und_Hluw | 1.71M | 374.92K | 2.09M | 70.77M | 15.47M | 86.25M | 3.19GB | 3.45GB | 6.64GB | Fineweb-2, New CC |
| ast_Latn | 1.63M | 144.18K | 1.77M | 213.12M | 19.08M | 232.20M | 1.39GB | 385.45MB | 1.78GB | Fineweb-2, MaLA, New CC |
| jpn_Japn | 1.60M | 177.40K | 1.78M | 148.77M | 17.99M | 166.76M | 6.05GB | 2.16GB | 8.21GB | MaLA |
| ibo_Latn | 1.59M | 117.64K | 1.71M | 233.50M | 16.65M | 250.14M | 1.45GB | 446.07MB | 1.89GB | Fineweb-2, MaLA |
| und_Grek | 1.57M | 224.66K | 1.79M | 755.84M | 108.19M | 864.02M | 6.94GB | 7.17GB | 14.12GB | Fineweb-2, New CC |
| mri_Latn | 1.53M | 133.72K | 1.67M | 354.50M | 28.72M | 383.22M | 1.71GB | 472.53MB | 2.18GB | Fineweb-2, MaLA |
| ars_Arab | 1.53M | 108.78K | 1.64M | 461.05M | 32.76M | 493.81M | 4.88GB | 1.85GB | 6.73GB | Fineweb-2, New CC |
| anp_Deva | 1.44M | 140.26K | 1.58M | 805.49M | 78.54M | 884.04M | 10.69GB | 2.12GB | 12.81GB | Fineweb-2, MaLA |
| khk_Cyrl | 1.44M | 128.14K | 1.57M | 615.04M | 54.80M | 669.84M | 8.17GB | 1.83GB | 10.00GB | Fineweb-2, New CC |
| und_Shrd | 1.41M | 216.59K | 1.62M | 130.80M | 20.13M | 150.93M | 6.06GB | 2.35GB | 8.40GB | Fineweb-2, New CC |
| lao_Laoo | 1.40M | 105.80K | 1.50M | 628.08M | 49.71M | 677.79M | 7.65GB | 1.36GB | 9.01GB | Fineweb-2, MaLA, New CC |
| und_Lina | 1.37M | 271.63K | 1.64M | 130.39M | 25.87M | 156.26M | 6.97GB | 3.85GB | 10.82GB | Fineweb-2, New CC |
| und_Samr | 1.35M | 158.99K | 1.51M | 64.06M | 7.54M | 71.59M | 4.30GB | 1.72GB | 6.02GB | Fineweb-2, New CC |
| ori_Orya | 1.34M | 145.91K | 1.48M | 128.69M | 11.97M | 140.66M | 2.16GB | 770.15MB | 2.93GB | MaLA |
| jav_Latn | 1.26M | 122.51K | 1.38M | 379.69M | 35.26M | 414.95M | 1.96GB | 587.75MB | 2.55GB | Fineweb-2, MaLA, New CC |
| yid_Hebr | 1.25M | 160.66K | 1.41M | 287.37M | 36.14M | 323.51M | 2.84GB | 1.30GB | 4.14GB | MaLA, New CC |
| und_Cans | 1.23M | 248.05K | 1.48M | 106.39M | 21.43M | 127.83M | 3.48GB | 2.77GB | 6.25GB | Fineweb-2, New CC |

Table 11: **Data Cleaning Statistics (part III):** Comparison of document count, token count, disk size, and sources before and after data cleaning in DCAD-2000.

| Lang Code | Documents | | | Tokens | | | Disk Size | | | Source |
|---|---|---|---|---|---|---|---|---|---|---|
| | keep | remove | total | keep | remove | total | keep | remove | total | |
| nya_Latn | 1.21M | 138.31K | 1.34M | 230.59M | 26.29M | 256.88M | 1.34GB | 437.81MB | 1.78GB | Fineweb-2, MaLA |
| hmn_Latn | 1.20M | 195.18K | 1.40M | 173.07M | 28.59M | 201.66M | 1.08GB | 543.90MB | 1.63GB | MaLA |
| tir_Ethi | 1.20M | 78.32K | 1.28M | 125.79M | 8.16M | 133.96M | 1.15GB | 290.56MB | 1.44GB | Fineweb-2, MaLA |
| uig_Arab | 1.19M | 78.60K | 1.27M | 513.72M | 37.42M | 551.15M | 3.72GB | 937.66MB | 4.65GB | Fineweb-2, MaLA, New CC |
| wln_Latn | 1.18M | 74.38K | 1.25M | 53.99M | 3.61M | 57.59M | 520.40MB | 78.21MB | 598.61MB | Fineweb-2, MaLA, New CC |
| und_Adlm | 1.12M | 194.29K | 1.32M | 43.63M | 7.55M | 51.18M | 1.10GB | 853.95MB | 1.95GB | Fineweb-2, New CC |
| und_Egyp | 1.12M | 190.50K | 1.31M | 97.41M | 16.58M | 113.99M | 2.54GB | 3.52GB | 6.05GB | Fineweb-2, New CC |
| und_Syrc | 1.12M | 115.88K | 1.23M | 42.71M | 4.43M | 47.14M | 20.68GB | 4.34GB | 25.01GB | Fineweb-2, New CC |
| swh_Latn | 1.12M | 82.67K | 1.20M | 449.92M | 32.71M | 482.63M | 3.34GB | 803.12MB | 4.15GB | Fineweb-2, MaLA |
| yor_Latn | 1.12M | 108.67K | 1.22M | 189.62M | 18.77M | 208.39M | 1.08GB | 304.95MB | 1.38GB | Fineweb-2, MaLA, New CC |
| uzn_Latn | 1.03M | 68.06K | 1.10M | 466.19M | 30.78M | 496.97M | 4.03GB | 1.03GB | 5.06GB | Fineweb-2, New CC |
| und_Mend | 1.03M | 293.72K | 1.32M | 16.58M | 4.75M | 21.33M | 893.39MB | 2.06GB | 2.95GB | Fineweb-2, New CC |
| xho_Latn | 1.02M | 88.44K | 1.11M | 168.59M | 13.93M | 182.52M | 1.19GB | 247.71MB | 1.44GB | Fineweb-2, MaLA |
| gla_Latn | 1.01M | 115.44K | 1.13M | 518.47M | 76.34M | 594.81M | 2.03GB | 904.76MB | 2.94GB | Fineweb-2, MaLA, New CC |
| bre_Latn | 980.75K | 86.36K | 1.07M | 134.68M | 11.68M | 146.37M | 757.53MB | 231.46MB | 988.99MB | Fineweb-2, MaLA, New CC |
| sot_Latn | 917.37K | 78.48K | 995.85K | 223.24M | 17.82M | 241.06M | 1.09GB | 283.15MB | 1.37GB | Fineweb-2, MaLA |
| nan_Latn | 905.48K | 86.68K | 992.16K | 26.58M | 2.54M | 29.12M | 483.99MB | 95.09MB | 579.08MB | Fineweb-2, MaLA |
| tel_Latn | 898.42K | 92.51K | 990.93K | 204.17M | 21.27M | 225.44M | 843.51MB | 444.92MB | 1.29GB | Fineweb-2, MaLA |
| bew_Latn | 885.97K | 99.33K | 985.30K | 370.27M | 41.51M | 411.78M | 2.85GB | 776.53MB | 3.62GB | Fineweb-2, New CC |
| smo_Latn | 883.15K | 83.25K | 966.41K | 241.45M | 21.17M | 262.62M | 1.15GB | 290.83MB | 1.44GB | Fineweb-2, MaLA |
| glk_Arab | 876.52K | 99.66K | 976.18K | 44.95M | 5.30M | 50.24M | 630.38MB | 171.44MB | 801.82MB | Fineweb-2, MaLA |
| che_Cyrl | 875.25K | 117.29K | 992.54K | 118.78M | 15.18M | 133.96M | 1.05GB | 346.83MB | 1.40GB | Fineweb-2, MaLA, New CC |
| orm_Latn | 859.55K | 77.40K | 936.95K | 35.46M | 3.19M | 38.65M | 476.68MB | 150.02MB | 626.69MB | MaLA |
| zho_Hani | 840.53K | 65.42K | 905.95K | 578.50M | 46.68M | 625.18M | 2.67GB | 980.93MB | 3.65GB | MaLA |
| haw_Latn | 808.97K | 88.12K | 897.10K | 227.68M | 23.61M | 251.29M | 869.19MB | 300.40MB | 1.17GB | Fineweb-2, MaLA |
| pnb_Arab | 806.70K | 71.03K | 877.73K | 133.55M | 11.76M | 145.31M | 881.83MB | 493.40MB | 1.38GB | Fineweb-2, MaLA, New CC |
| oci_Latn | 760.65K | 59.16K | 819.82K | 123.30M | 10.54M | 133.84M | 706.68MB | 193.69MB | 900.37MB | Fineweb-2, MaLA, New CC |
| und_Linb | 735.07K | 107.67K | 842.75K | 52.97M | 7.76M | 60.73M | 6.30GB | 997.90MB | 7.30GB | Fineweb-2, New CC |
| chv_Cyrl | 731.68K | 60.72K | 792.40K | 188.93M | 16.35M | 205.28M | 1.10GB | 361.84MB | 1.46GB | Fineweb-2, MaLA, New CC |
| kin_Latn | 701.70K | 67.29K | 768.99K | 197.65M | 16.84M | 214.49M | 1.43GB | 160.27MB | 1.59GB | Fineweb-2, MaLA |
| srp_Latn | 630.88K | 54.65K | 685.53K | 158.44M | 13.19M | 171.63M | 775.01MB | 209.48MB | 984.49MB | MaLA |
| und_Brai | 590.10K | 125.33K | 715.43K | 57.85M | 12.29M | 70.13M | 1.94GB | 1.30GB | 3.24GB | Fineweb-2, New CC |
| kaa_Cyrl | 588.71K | 48.01K | 636.72K | 1.08B | 86.21M | 1.16B | 3.58GB | 620.59MB | 4.20GB | Fineweb-2, New CC |
| lug_Latn | 570.88K | 40.31K | 611.19K | 36.43M | 2.65M | 39.08M | 344.92MB | 85.21MB | 430.13MB | Fineweb-2, MaLA |
| und_Sgnw | 567.29K | 106.45K | 673.74K | 37.34M | 7.01M | 44.34M | 1.40GB | 1.11GB | 2.50GB | Fineweb-2, New CC |
| pcm_Latn | 563.55K | 80.45K | 644.00K | 135.97M | 19.60M | 155.57M | 1.45GB | 231.26MB | 1.68GB | Fineweb-2, MaLA |
| pbt_Arab | 556.45K | 36.70K | 593.15K | 273.04M | 18.00M | 291.04M | 2.40GB | 481.43MB | 2.88GB | Fineweb-2, MaLA |
| min_Latn | 548.22K | 32.98K | 581.19K | 28.26M | 1.78M | 30.04M | 326.92MB | 43.32MB | 370.24MB | Fineweb-2, MaLA |
| tuk_Latn | 526.60K | 48.40K | 575.00K | 211.69M | 23.04M | 234.74M | 1.14GB | 368.23MB | 1.51GB | Fineweb-2, MaLA |
| lim_Latn | 526.45K | 43.83K | 570.28K | 49.16M | 4.85M | 54.01M | 338.07MB | 70.26MB | 408.33MB | Fineweb-2, MaLA, New CC |
| und_Hung | 520.10K | 155.23K | 675.33K | 42.34M | 12.64M | 54.98M | 1.94GB | 2.32GB | 4.25GB | Fineweb-2, New CC |
| gsw_Latn | 519.60K | 64.76K | 584.36K | 171.13M | 22.15M | 193.28M | 2.02GB | 248.45MB | 2.27GB | Fineweb-2, MaLA, New CC |
| aze_Arab | 481.85K | 107.19K | 589.05K | 16.65M | 3.70M | 20.35M | 283.94MB | 125.40MB | 409.33MB | MaLA |
| kmr_Latn | 473.75K | 37.03K | 510.79K | 239.78M | 19.24M | 259.01M | 1.64GB | 366.13MB | 2.01GB | Fineweb-2, MaLA, New CC |
| roh_Latn | 467.79K | 40.88K | 508.66K | 59.96M | 5.00M | 64.96M | 373.84MB | 133.62MB | 507.46MB | Fineweb-2, New CC |
| vec_Latn | 451.53K | 28.94K | 480.47K | 35.51M | 2.41M | 37.92M | 248.96MB | 70.25MB | 319.21MB | Fineweb-2, MaLA |
| san_Deva | 426.60K | 30.30K | 456.90K | 186.19M | 14.19M | 200.38M | 1.37GB | 884.42MB | 2.25GB | Fineweb-2, MaLA, New CC |
| und_Bali | 422.49K | 77.08K | 499.57K | 39.62M | 7.23M | 46.84M | 1.19GB | 662.91MB | 1.85GB | Fineweb-2, New CC |
| und_Nshu | 419.71K | 89.40K | 509.11K | 38.53M | 8.21M | 46.74M | 993.06MB | 1.28GB | 2.27GB | Fineweb-2, New CC |
| und_Modi | 386.82K | 67.33K | 454.15K | 52.58M | 9.15M | 61.73M | 16.45GB | 7.42GB | 23.87GB | Fineweb-2, New CC |
| gmh_Latn | 383.58K | 47.47K | 431.05K | 769.12M | 95.18M | 864.30M | 5.51GB | 1.42GB | 6.93GB | Fineweb-2, New CC |
| sco_Latn | 382.19K | 37.49K | 419.69K | 43.05M | 4.46M | 47.52M | 357.63MB | 98.46MB | 456.10MB | Fineweb-2, MaLA |
| nds_Latn | 379.54K | 44.24K | 423.78K | 79.45M | 11.68M | 91.13M | 384.74MB | 126.48MB | 511.22MB | Fineweb-2, MaLA, New CC |
| und_Lana | 377.58K | 110.80K | 488.38K | 47.55M | 13.95M | 61.50M | 688.16MB | 2.05GB | 2.74GB | Fineweb-2, New CC |
| azb_Arab | 376.14K | 24.16K | 400.30K | 81.10M | 6.51M | 87.61M | 615.69MB | 203.89MB | 819.58MB | Fineweb-2, MaLA, New CC |
| tsn_Latn | 375.82K | 23.43K | 399.25K | 24.79M | 1.54M | 26.33M | 206.56MB | 41.32MB | 247.88MB | Fineweb-2, MaLA |
| und_Mong | 364.92K | 51.36K | 416.28K | 78.04M | 10.98M | 89.03M | 1.32GB | 827.40MB | 2.15GB | Fineweb-2, New CC |
| sah_Cyrl | 357.02K | 24.17K | 381.19K | 110.13M | 7.77M | 117.89M | 1.05GB | 202.76MB | 1.25GB | MaLA, New CC |
| und_Ethi | 351.77K | 49.20K | 400.97K | 39.75M | 5.56M | 45.31M | 1.23GB | 1.08GB | 2.31GB | Fineweb-2, New CC |
| rus_Latn | 349.61K | 47.55K | 397.17K | 77.31M | 10.54M | 87.85M | 755.00MB | 485.49MB | 1.24GB | MaLA |
| pri_Latn | 348.99K | 27.20K | 376.20K | 142.27M | 11.09M | 153.36M | 2.15GB | 505.82MB | 2.66GB | Fineweb-2, New CC |
| und_Hebr | 345.20K | 46.87K | 392.07K | 17.42M | 2.36M | 19.78M | 548.23MB | 461.10MB | 1.01GB | Fineweb-2, New CC |
| mon_Latn | 344.80K | 46.68K | 391.48K | 31.56M | 4.27M | 35.84M | 180.12MB | 271.24MB | 451.37MB | MaLA |
| pap_Latn | 339.80K | 22.62K | 362.42K | 127.89M | 8.52M | 136.41M | 678.73MB | 223.10MB | 901.83MB | Fineweb-2, MaLA |
| tgk_Latn | 337.95K | 48.39K | 386.35K | 26.44M | 3.79M | 30.22M | 198.08MB | 219.19MB | 417.27MB | MaLA |
| plt_Latn | 330.57K | 28.23K | 358.80K | 118.46M | 8.02M | 126.48M | 951.31MB | 189.98MB | 1.14GB | Fineweb-2, MaLA |
| lmo_Latn | 324.18K | 29.25K | 353.43K | 41.37M | 4.09M | 45.46M | 230.80MB | 58.92MB | 289.72MB | Fineweb-2, MaLA, New CC |
| bod_Tibt | 318.52K | 34.22K | 352.75K | 252.06M | 28.33M | 280.39M | 3.37GB | 998.77MB | 4.37GB | MaLA, New CC |
| und_Saur | 315.78K | 73.82K | 389.60K | 15.26M | 3.57M | 18.83M | 398.55MB | 489.07MB | 887.62MB | Fineweb-2, New CC |
| yue_Hani | 300.49K | 34.04K | 334.53K | 9.02M | 1.03M | 10.06M | 790.86MB | 161.03MB | 951.90MB | Fineweb-2, MaLA, New CC |

Table 12: **Data Cleaning Statistics (part IV):** Comparison of document count, token count, disk size, and sources before and after data cleaning in DCAD-2000.

| Lang Code | Documents | | | Tokens | | | Disk Size | | | Source |
|---|---|---|---|---|---|---|---|---|---|---|
| | keep | remove | total | keep | remove | total | keep | remove | total | |
| bar_Latn | 270.31K | 30.79K | 301.10K | 92.48M | 12.46M | 104.94M | 318.42MB | 142.36MB | 460.78MB | Fineweb-2, MaLA |
| und_Thaa | 262.00K | 37.56K | 299.56K | 7.68M | 1.10M | 8.78M | 391.60MB | 263.21MB | 654.81MB | Fineweb-2, New CC |
| und_Dupl | 258.90K | 53.06K | 311.96K | 14.14M | 2.90M | 17.04M | 752.58MB | 502.95MB | 1.26GB | Fineweb-2, New CC |
| arg_Latn | 258.20K | 22.52K | 280.72K | 29.97M | 3.05M | 33.02M | 207.88MB | 43.58MB | 251.45MB | Fineweb-2, MaLA, New CC |
| pms_Latn | 258.13K | 20.25K | 278.38K | 23.55M | 1.86M | 25.41M | 172.17MB | 39.05MB | 211.22MB | Fineweb-2, MaLA, New CC |
| hif_Latn | 254.95K | 37.47K | 292.41K | 220.19M | 38.74M | 258.93M | 779.02MB | 879.71MB | 1.66GB | Fineweb-2, MaLA |
| und_Thai | 254.35K | 47.64K | 301.99K | 47.88M | 8.97M | 56.85M | 868.70MB | 325.83MB | 1.19GB | Fineweb-2, New CC |
| und_Runr | 252.18K | 39.00K | 291.18K | 154.68M | 23.92M | 178.61M | 1.25GB | 3.28GB | 4.52GB | Fineweb-2, New CC |
| und_Vaii | 243.47K | 93.27K | 336.73K | 71.28M | 27.31M | 98.59M | 513.30MB | 1.88GB | 2.39GB | Fineweb-2, New CC |
| vol_Latn | 241.22K | 23.73K | 264.95K | 12.26M | 1.28M | 13.54M | 126.45MB | 27.79MB | 154.24MB | Fineweb-2, MaLA, New CC |
| und_Glag | 237.68K | 72.07K | 309.75K | 20.38M | 6.18M | 26.56M | 476.61MB | 951.96MB | 1.43GB | Fineweb-2, New CC |
| nrm_Latn | 234.99K | 31.99K | 266.97K | 71.12M | 9.68M | 80.80M | 654.26MB | 233.97MB | 888.23MB | Fineweb-2, MaLA |
| aeb_Arab | 230.69K | 32.19K | 262.88K | 51.79M | 7.23M | 59.01M | 641.42MB | 232.91MB | 874.33MB | Fineweb-2, New CC |
| kat_Latn | 229.64K | 46.98K | 276.62K | 37.42M | 7.66M | 45.08M | 247.34MB | 365.42MB | 612.76MB | MaLA |
| ido_Latn | 222.87K | 22.62K | 245.49K | 15.65M | 1.48M | 17.13M | 131.86MB | 35.81MB | 167.67MB | Fineweb-2, MaLA, New CC |
| kal_Latn | 220.32K | 17.35K | 237.67K | 76.08M | 6.03M | 82.11M | 371.13MB | 202.28MB | 573.42MB | Fineweb-2, MaLA |
| pam_Latn | 219.65K | 22.53K | 242.18K | 21.42M | 2.45M | 23.87M | 129.69MB | 37.16MB | 166.84MB | Fineweb-2, MaLA |
| und_Khmr | 216.99K | 36.25K | 253.24K | 10.97M | 1.83M | 12.80M | 473.35MB | 417.98MB | 891.34MB | Fineweb-2, New CC |
| lus_Latn | 206.91K | 16.42K | 223.33K | 66.59M | 5.16M | 71.75M | 387.16MB | 114.21MB | 501.37MB | Fineweb-2, MaLA |
| und_Mymr | 204.74K | 27.30K | 232.03K | 5.63M | 751.18K | 6.39M | 283.14MB | 249.17MB | 532.31MB | Fineweb-2, New CC |
| und_Tibt | 201.49K | 32.84K | 234.33K | 15.44M | 2.52M | 17.95M | 970.52MB | 505.25MB | 1.48GB | Fineweb-2, New CC |
| und_Dsrt | 198.00K | 37.90K | 235.90K | 4.47M | 855.49K | 5.32M | 248.83MB | 562.92MB | 811.75MB | Fineweb-2, New CC |
| und_Geor | 196.35K | 49.50K | 245.85K | 22.22M | 5.60M | 27.83M | 374.55MB | 629.81MB | 1.00GB | Fineweb-2, New CC |
| new_Deva | 187.27K | 16.23K | 203.49K | 23.86M | 2.07M | 25.93M | 302.85MB | 89.56MB | 392.41MB | Fineweb-2, MaLA, New CC |
| und_Mroo | 186.14K | 22.85K | 208.99K | 6.42M | 788.69K | 7.21M | 2.43GB | 335.38MB | 2.77GB | Fineweb-2, New CC |
| sme_Latn | 184.43K | 14.88K | 199.30K | 42.27M | 3.53M | 45.80M | 318.80MB | 92.35MB | 411.15MB | Fineweb-2, MaLA |
| und_Bopo | 181.71K | 24.45K | 206.16K | 30.63M | 4.12M | 34.75M | 3.45GB | 890.68MB | 4.35GB | Fineweb-2, New CC |
| nso_Latn | 175.98K | 9.66K | 185.64K | 18.89M | 1.08M | 19.97M | 111.81MB | 32.30MB | 144.10MB | Fineweb-2, MaLA |
| und_Armn | 168.06K | 46.69K | 214.75K | 33.05M | 9.18M | 42.24M | 347.17MB | 515.75MB | 862.92MB | Fineweb-2, New CC |
| und_Mtei | 166.92K | 19.64K | 186.57K | 48.49M | 5.71M | 54.20M | 795.85MB | 570.91MB | 1.37GB | Fineweb-2, New CC |
| scn_Latn | 162.55K | 10.71K | 173.26K | 18.07M | 1.48M | 19.55M | 125.29MB | 25.45MB | 150.75MB | Fineweb-2, MaLA, New CC |
| ina_Latn | 159.80K | 16.99K | 176.79K | 13.62M | 1.49M | 15.11M | 104.31MB | 28.02MB | 132.33MB | Fineweb-2, MaLA, New CC |
| lld_Latn | 154.22K | 24.98K | 179.20K | 8.00M | 1.25M | 9.25M | 90.50MB | 16.81MB | 107.31MB | Fineweb-2, MaLA |
| und_Khar | 153.37K | 40.04K | 193.41K | 6.75M | 1.76M | 8.52M | 250.30MB | 182.38MB | 432.67MB | Fineweb-2, New CC |
| hyw_Armn | 142.71K | 12.89K | 155.60K | 60.77M | 5.52M | 66.29M | 863.69MB | 174.18MB | 1.04GB | Fineweb-2, MaLA |
| und_Deva | 141.13K | 26.07K | 167.20K | 35.96M | 6.64M | 42.60M | 255.99MB | 1.53GB | 1.79GB | Fineweb-2, New CC |
| abk_Cyrl | 139.72K | 12.86K | 152.58K | 7.73M | 671.84K | 8.40M | 100.31MB | 14.57MB | 114.88MB | Fineweb-2, MaLA |
| und_Brah | 138.03K | 22.72K | 160.75K | 7.85M | 1.29M | 9.15M | 273.71MB | 243.75MB | 517.47MB | Fineweb-2, New CC |
| bpy_Beng | 135.66K | 9.50K | 145.16K | 9.30M | 766.18K | 10.07M | 141.90MB | 28.27MB | 170.17MB | Fineweb-2, MaLA, New CC |
| bew_Cyrl | 133.83K | 13.49K | 147.32K | 3.37M | 339.68K | 3.71M | 74.12MB | 15.81MB | 89.93MB | MaLA |
| lin_Latn | 133.64K | 8.68K | 142.32K | 16.04M | 1.37M | 17.41M | 115.63MB | 32.27MB | 147.89MB | Fineweb-2, MaLA |
| und_Bhks | 131.90K | 27.03K | 158.93K | 3.93M | 805.58K | 4.74M | 190.96MB | 154.63MB | 345.59MB | Fineweb-2, New CC |
| oss_Cyrl | 128.06K | 13.97K | 142.03K | 84.80M | 9.56M | 94.36M | 390.43MB | 167.02MB | 557.45MB | Fineweb-2, MaLA, New CC |
| tgk_Arab | 127.77K | 14.97K | 142.75K | 11.61M | 1.36M | 12.97M | 104.11MB | 55.51MB | 159.62MB | MaLA |
| szl_Latn | 127.60K | 10.33K | 137.93K | 8.52M | 738.21K | 9.25M | 89.99MB | 12.81MB | 102.80MB | Fineweb-2, MaLA |
| mww_Latn | 122.30K | 10.22K | 132.52K | 98.37M | 8.22M | 106.59M | 536.48MB | 104.20MB | 640.68MB | Fineweb-2, New CC |
| sdh_Arab | 120.04K | 14.20K | 134.24K | 35.26M | 4.50M | 39.76M | 466.52MB | 136.99MB | 603.52MB | Fineweb-2, MaLA |
| und_Hmnp | 118.87K | 12.33K | 131.20K | 6.83M | 708.37K | 7.54M | 436.28MB | 151.81MB | 588.09MB | Fineweb-2, New CC |
| srd_Latn | 118.78K | 8.14K | 126.92K | 15.38M | 1.23M | 16.61M | 119.77MB | 24.18MB | 143.95MB | Fineweb-2, MaLA |
| mhr_Cyrl | 118.77K | 12.58K | 131.35K | 30.71M | 3.17M | 33.88M | 278.82MB | 75.27MB | 354.09MB | Fineweb-2, MaLA, New CC |
| ydd_Hebr | 117.78K | 7.28K | 125.06K | 73.71M | 4.55M | 78.26M | 879.66MB | 120.71MB | 1.00GB | Fineweb-2, MaLA |
| diq_Latn | 117.09K | 11.78K | 128.87K | 9.75M | 962.88K | 10.71M | 75.44MB | 16.34MB | 91.79MB | Fineweb-2, MaLA, New CC |
| und_Telu | 115.91K | 30.83K | 146.74K | 9.00M | 2.39M | 11.40M | 409.30MB | 426.99MB | 836.29MB | Fineweb-2, New CC |
| que_Latn | 114.28K | 23.93K | 138.21K | 4.28M | 896.83K | 5.18M | 57.76MB | 33.84MB | 91.59MB | MaLA, New CC |
| run_Latn | 114.03K | 9.29K | 123.32K | 24.63M | 1.97M | 26.60M | 218.56MB | 39.33MB | 257.89MB | Fineweb-2, New CC |
| hsb_Latn | 112.76K | 9.95K | 122.71K | 25.10M | 2.04M | 27.14M | 153.09MB | 23.81MB | 176.90MB | Fineweb-2, MaLA, New CC |
| wol_Latn | 108.94K | 11.08K | 120.02K | 11.76M | 1.37M | 13.13M | 95.99MB | 29.75MB | 125.74MB | Fineweb-2, MaLA |
| rmy_Latn | 108.23K | 21.11K | 129.34K | 284.56M | 55.71M | 340.26M | 2.54GB | 98.95MB | 2.64GB | Fineweb-2, MaLA |
| und_Phag | 107.75K | 17.58K | 125.34K | 3.41M | 556.36K | 3.97M | 141.68MB | 93.31MB | 234.99MB | Fineweb-2, New CC |
| und_Merc | 107.52K | 38.04K | 145.56K | 7.61M | 2.69M | 10.30M | 215.43MB | 472.23MB | 687.66MB | Fineweb-2, New CC |
| urd_Latn | 106.75K | 12.60K | 119.35K | 139.19M | 16.43M | 155.63M | 312.70MB | 140.28MB | 452.98MB | Fineweb-2, New CC |
| kiu_Latn | 106.48K | 10.36K | 116.84K | 36.53M | 3.76M | 40.29M | 289.67MB | 193.73MB | 483.39MB | Fineweb-2, MaLA |
| cak_Latn | 106.28K | 6.64K | 112.92K | 6.08M | 438.75K | 6.52M | 66.17MB | 10.86MB | 77.03MB | Fineweb-2, MaLA |
| ilo_Latn | 106.18K | 7.83K | 114.01K | 28.61M | 2.06M | 30.67M | 143.69MB | 37.61MB | 181.30MB | Fineweb-2, MaLA, New CC |
| und_Kali | 105.87K | 24.33K | 130.19K | 1.39M | 318.99K | 1.71M | 105.22MB | 91.44MB | 196.66MB | Fineweb-2, New CC |
| und_Plrd | 104.31K | 21.07K | 125.38K | 5.47M | 1.10M | 6.57M | 214.53MB | 225.25MB | 439.77MB | Fineweb-2, New CC |
| und_Orya | 104.03K | 26.52K | 130.56K | 10.14M | 2.59M | 12.73M | 299.43MB | 387.64MB | 687.07MB | Fineweb-2, New CC |
| und_Lisu | 101.47K | 20.05K | 121.52K | 24.00M | 4.74M | 28.74M | 204.23MB | 527.19MB | 731.42MB | Fineweb-2, New CC |
| und_Hmng | 101.02K | 23.34K | 124.36K | 5.37M | 1.24M | 6.61M | 153.20MB | 196.99MB | 350.19MB | Fineweb-2, New CC |

