# OpenReview forum: "DCAD-2000: A Multilingual Dataset across 2000+ Languages with Data Cleaning as Anomaly Detection"
_NeurIPS.cc/2025/Datasets_and_Benchmarks_Track — NeurIPS 2025 Datasets and Benchmarks Track poster_

### Official Review · Reviewer_b7Zx · 2025-06-28

**Rating:** 4
**Confidence:** 4

**Summary:**

This paper introduces a new multilingual DCAD-2000 dataset that combines existing datasets with recent CommonCrawl data and covers over two thousand high-, medium, low-, and very low-resource languages and more than eight billion documents. The authors propose an anomaly detection based document cleaning approach to filter spam or noisy documents based on eight quality metrics and conduct extensive experiments for comparing data quality.

**Additional Feedback:**

- The Ksyun server link is not working.

**Dataset Code Accessibility:**

Yes

**Dataset Code Comments:**

The dataset and code are available via GitHib and HuggingFace Datasets repository.

**Ethical Considerations:**

No, there are no or only very minor ethics concerns

**Final Justification:**

While I appreciate the authors' response, my minor concerns remain.

**Limitations Weaknesses:**

- The authors use anomaly detection to filter low quality documents. "Data selection" would be a more appropriate term for this approach than "data cleaning". Data cleaning traditionally refers to fixing data quality issues.

- A main claim of this paper is that the anomaly detection methods dynamically identify low-quality documents in contrast to traditional methods that use fixed thresholds. However, the proposed approach still requires some threshold-like hyperparameters such as the `contamination` parameter in the isolation forest (https://scikit-learn.org/stable/modules/generated/sklearn.ensemble.IsolationForest.html)

- The score improvements of the datasets (Figure 4) and the accuracy improvements of the anomaly detection method over other approaches (Figure 3) seem to diminish with larger models. Need better explanations.

- The difference between DCAD-2000 and Fineweb-2 is unclear. How important are low- and very low-resource languages that DCAD-2000 better covers?

- What is the exact size of each dataset (number of documents, which languages are covered)?

- Regarding the data collection for DCAD-2000, it would be interesting to see if there is an intersection between FineWeb, FineWeb-2, MaLA, and the new CC-2024, especially since FineWeb is missing in Table 1. Moreover, it is not clear how Fineweb-2 preparation is comparable with your preparation or MaLA (deduplication via MinHashLSH vs Datatrove *Fineweb-2 fine-tunes thresholds for more than 1,000 languages, this process is computationally intensive and time-consuming*) Finally, are there duplicates between datasets, and how did authors filter those?

- There are a few points that are not clear about data cleaning. (1) Do authors remove documents from low-resource languages if there are only a few documents? How is it decided? (2) How many documents were filtered from each dataset? What did those documents contain? How did authors ensure that filtered documents are not false positives?

- The anomaly detection methods are computationally expensive compared to the threshold-based approaches. The paper will benefit from more details regarding compute cost and execution time comparison across cleaning approaches.

- It is unclear how do authors evaluate very low-resource languages, since FineTask covers only nine languages. Moreover, for all three evaluated models, performance in Telugu (a less common language) is significantly low, which requires better explanations.

- *Insufficient data cleaning* point in line 29 requires better references. The cited papers are about rare Chinese (low) languages, and the datasets are older than Fineweb-2 datasets.

- The work would benefit from more extensive related work (differences between each dataset, more about exact data cleaning/anomaly detection methods).

**Strengths Contributions:**

- A new multilingual dataset that includes over 2,282 languages and 8.63 billion documents to improve the multilingual capabilities of LLMs. The authors carefully selected four data sources and applied deduplication as a pre-processing.

- A data cleaning pipeline that, in some cases, substantially improves the downstream task performance. Re-thinking document quality, especially for less-represented languages, is an important line of research.

- Taking into account advancements of LLMs, the general community would benefit from a dataset containing filtered documents covering not only high-resource languages but also a variety of medium- and low-resource languages.

- The paper is generally well-written.

---

> ### Author Rebuttal · Authors · 2025-07-31
>
> We sincerely thank the reviewer for the thoughtful and detailed feedback. Due to character limit restrictions, we have streamlined the text of your question and are providing the corresponding response below:
>
> > **Q1:** Use "data selection" rather than "data cleaning".
>
> **A1:** Thank you for your valuable feedback. We agree that in a broad sense, data cleaning can indeed be viewed as a form of "data selection," and we appreciate your suggestion. We will consider adopting the term "data selection" in the next version of the paper to more accurately reflect the approach we used. However, we also want to clarify that our method focuses on identifying and removing low-quality documents, which aligns with the general practice of data cleaning, albeit with a different technique.
>
> ***
>
> > **Q2:** The approach still relies on threshold-like hyperparameters.
>
> **A2:** Thank you for pointing this out. While it is true that anomaly detection methods, including Isolation Forest, require certain hyperparameters, such as the `contamination` parameter, the key distinction lies in how these parameters/threshold are used. Unlike traditional heuristic-based methods, which require the adjustment of thresholds (eight features) for different languages, the hyperparameters in anomaly detection only need to be fine-tuned once. Once set, the method applies consistently across all languages without the need for further adjustments. This flexibility ensures that our approach is more scalable and less resource-intensive compared to fixed-threshold methods.
>
> ***
>
> > **Q3:** Performance gains become diminish with larger models.
>
> **A3:** Thank you for your comment. This observation aligns with findings in existing literatures [1-3], where larger models tend to be more robust to noise. Several studies have shown that models with more parameters exhibit improved generalization capabilities, which can lead to diminished performance improvements from data cleaning or anomaly detection methods as the model size increases. Larger models can learn to filter out noise during training, reducing the necessity for pre-processing steps. We will add a clearer explanation of this effect in the revised version, citing relevant works to provide further context.
>
> **References**
>
> [1] Scaling Laws for Neural Language Models (Kaplan et al., arXiv 2020).
>
> [2] Palm: Scaling Language Modeling with Pathways (Chowdhery et al., JMLR 2022).
>
> [3] Scaling Trends in Language Model Robustness (Howe et al., ICML 2025).
> ***
>
> > **Q4:** Compare DCAD-2000 with Fineweb-2.
>
> **A4:** Thank you for your comment. We provide a broad comparison between DCAD-2000 and Fineweb-2 in **Table 1,** where DCAD-2000 demonstrates a stronger advantage in the coverage of medium-resource languages. For a more detailed comparison of low- and very low-resource languages, please refer to **Table 3**.
>
> In addition, our experiments show that models trained on DCAD-2000 outperform those trained on Fineweb-2 across high, medium, low, and very low-resource languages in several multilingual benchmarks. This highlights the significance of low- and very low-resource languages in improving model performance, particularly for underrepresented languages, which are better covered by DCAD-2000.
> ***
>
> > **Q5:** Exact size of each dataset.
>
> **A5:** Thank you for your question. Detailed statistics for all high- and medium-resource languages in DCAD-2000, including the total number of documents, tokens, dataset size, and data sources, are provided in **Tables 6-9** in the appendix. However, due to the extensive coverage of over 2,000 languages, we do not present the complete statistics for all languages in the main paper. Instead, we have made this information available on HuggingFace and GitHub, where you can find the full dataset details, including the number of documents and languages covered.
>
> ***
>
> > **Q6:** Duplication issues.
>
> **A6:** Thank you for your questions. Our responses to each point are as follows:
>
> + FineWeb is not included in Table 1 because it is an English-only dataset, whereas DCAD-2000 focuses on multilingual data. FineWeb-2 is the multilingual version of FineWeb and is included in the comparison.
>
> + As explained in **Section 3.1**, DCAD-2000 is built using existing datasets (MaLA, FineWeb, FineWeb-2) and newly extracted data from Common Crawl (New CC-2024). We further clean the existing datasets using our data cleaning framework to improve their quality.
>
> + We elaborate on our data preparation strategy in lines 98-101 of Section 3.1. In our approach, we prioritize multilingual coverage, data quality, and freshness. FineWeb-2 includes much of the Common Crawl data, while our newly extracted CC data comes from snapshots not covered by FineWeb. We also perform deduplication across datasets using our framework.
> - Yes, we conduct deduplication across datasets to ensure there is no overlap. This is done through a careful process of cross-dataset checking to avoid duplicate content and maintain data integrity.
> ****
>
> > **Q7:** Low-resource languages in DCAD.
>
> **A7:** Thank you for your thoughtful questions. We address each point below:
>
> (1) **Filtering Low-Resource Languages**: We do not specifically filter out documents from low-resource languages based on their quantity. Our data cleaning process focuses on improving data quality, not on the number of documents available in each language. Therefore, low-resource languages are not treated differently based on the number of documents they have
>
> (2) **Filtered Documents and Their Contents**: We provide detailed statistics on the data before and after cleaning in **Tables 6-9.** These tables include comparisons of the number of documents, tokens, and dataset size before and after cleaning. All this information, along with the full dataset, is publicly available on Hugging Face. We will also include a more detailed **case study** of the deleted and retained documents in the next version of the paper to provide a better understanding of the contents and the rationale behind filtering.
>
> ***
>
> > **Q8:** Computational considerations.
>
> **A8:** Thank you for your comment. It is true that our anomaly detection method requires more computational resources compared to traditional threshold-based approaches. However, the traditional methods also **require significant computation** for selecting different feature thresholds for each language, which can be computationally expensive.
>
> Overall, the additional computational cost in our method comes from the classification step of the anomaly detection algorithm. For example, when processing 1 million documents using Isolation Forest, our approach incurs an additional **2-5 minutes** of execution time. The exact time also depends on hardware resources, such as the number of CPU cores and available memory. We will add more detailed information about the computational costs and execution time comparisons in the next version of the paper.
> ***
>
> > **Q9:** Evaluation on very low-resource languages.
>
> **A9:** Thank you for your observations. Please see the response below:
>
> - **Evaluation of Very Low-Resource Languages**: Due to computational limitations, we evaluated the effectiveness of our framework using FineTask, which only covers nine languages, as shown in **Figure 3**. While this is a limited evaluation, it was necessary to demonstrate the general effectiveness of our proposed framework in a manageable setting. For very low-resource languages, we conducted additional evaluations across various multilingual benchmarks, as shown in **Table 3**. Our results indicate that the quality of the dataset after DCAD cleaning is indeed strong, even for very low-resource languages, which reinforces the effectiveness of our data cleaning approach.
> - **Performance in Telugu**: Telugu, being a low-resource language, has not been well-trained in the base models, which explains the relatively low performance observed in all three evaluated models. This outcome is expected given the limited training data available for Telugu in these models.
> ***
>
> > **Q10:** Better references.
>
> **A10:** Thank you for your comment. The paper by Sailor [18] specifically addresses datasets for a large number of Southeast Asian languages and provides a detailed analysis of the residual noise in previously cleaned datasets. While it is true that Sailor’s dataset predates Fineweb-2, we referenced it to highlight that even cleaned datasets may still contain substantial noise. This suggests that further cleaning is often necessary before training large language models, which is why we emphasize the importance of the "ready-for-training" field in Table 1.
>
> ***
>
> > **Q11:** Reference issues.
>
> **A11:** Thank you for your valuable suggestion. We will expand the related work section in the next version of the paper to provide a more detailed comparison of the datasets and a deeper explanation of the data cleaning and anomaly detection methods. This will help clarify the distinctions between datasets and offer a better understanding of our proposed approach.
>
> ***
>
> > **Q12:** The Ksyun server link is not working.
>
> **A12:** Thank you for flagging this. We acknowledge that the Ksyun Kubernetes platform link was related to our internal infrastructure setup and is not publicly accessible. In our experiments, we used a managed **Kubernetes (k8s)** cluster for orchestration and scalability. Kubernetes is an open-source system for automating the deployment, scaling, and management of containerized applications ([see official documentation](https://kubernetes.io/docs/concepts/overview/)). We will update the text to clarify this and remove the non-functional link in the final version.

---

> > ### Comment · Reviewer_b7Zx · 2025-08-04
> >
> > I would like to thank the authors for their response. While they clarified most points, the minor concerns remain. Therefore, I will keep my rating as is.

---

> > > ### Author Response · Authors · 2025-08-06
> > >
> > > Thank you for your response and for recognizing the clarifications we have made. Although the remaining concerns were not specified, we will make sure to revisit all aspects of the paper to further refine and clarify our contributions in the final version. We truly appreciate your feedback and the time you’ve dedicated to reviewing our work.

---

### Official Review · Reviewer_AMsz · 2025-07-01

**Rating:** 5
**Confidence:** 4

**Summary:**

The paper introduces **DCAD-2000**, a large-scale multilingual corpus spanning 2,282 languages and 159 scripts, constructed from **Common Crawl (CC-MAIN-2024-46)** and existing datasets **(MaLA, Fineweb-2)**. The core contribution reframes data cleaning as an anomaly detection task, using eight statistical features (e.g., word count, repetition ratios) and uses techniques from anomaly detection literature like **Isolation Forest** to remove anomalous documents. The authors demonstrate that DCAD-2000 outperforms existing corpora (e.g., Fineweb-2, MaLA) in multilingual benchmarks like SIB-200, FLORES-200, FineTask benchmark when used for continual pre-training of LLMs **(LLaMA-3.2-1B, Qwen-2.5-7B, Aya-expanse-32B)**. They also demonstrate that anomaly detection outperforms heuristic filtering in terms of Normalized accuracy on the FineTask benchmark.

**Dataset Code Accessibility:**

Yes

**Ethical Considerations:**

No, there are no or only very minor ethics concerns

**Final Justification:**

Thank you to the authors for sharing the additional evaluation results, cost-benefit analysis, and preliminary agreement statistics. These additions significantly strengthen the validation of the presented results. I trust that these key details - including the cost-benefit analysis, the planned full-scale evaluation on Global MMLU and INCLUDE, and the complete agreement statistics / qualitative analysis - will be thoroughly integrated into the final version of the manuscript or appendix.

As an additional suggestion, I recommend explicitly mentioning the name of the benchmark in the captions of Figures and Tables (e.g., Figure 3, Table 2), as this would help readers quickly grasp the context at a glance without needing to refer back to the main text.

Overall, the presented additional results, along with the detailed responses to other reviewers, strengthens the work considerably. I will update my score to reflect my positive assessment.

**Limitations Weaknesses:**

- **Confounded Benchmark Comparisons**:
DCAD-2000 subsumes data from Fineweb-2, MaLA, and New CC [Table 4]. When comparing against these "subsets" (e.g., Fineweb-2 in Table 3), superior performance may stem from broader data coverage rather than the cleaning pipeline. The experiments fail to isolate the impact of anomaly detection versus expanded data diversity and coverage. This undermines claims about the cleaning framework's efficacy [Table 3, Figure 4].

- **Inadequate Evaluation of Cleaning Pipeline**:
-  The anomaly detection method is evaluated only on FineTask [Figure 3], while dataset comparisons use multiple benchmarks (SIB-200, Glot500-c, FLORES-200) alongside FineTask benchmark. This discrepancy lacks justification. Authors should make their benchmarking consistent. Standard high-quality multilingual benchmarks like [Global MMLU](https://arxiv.org/abs/2412.03304) and [Include](https://arxiv.org/abs/2411.19799) and are omitted, authors should report the performance of DCAD on these standard benchmarks as well.

- **Baseline Rigor**:
Heuristic thresholds are tested superficially (e.g., "threshold-based filtering" in Figure 3) without exploring stricter or optimized thresholds. A cost-benefit analysis of anomaly detection versus refined heuristics is absent. Experiments conclusively demonstrating DCAD's superiority over rigorous threshold-based filtering are needed.

- **Qualitative Examples and Additional Statistics**
Authors should include few qualitative examples to  illustrate documents filtered by DCAD but retained by heuristics (or vice versa). Statistics on decision agreement (keep / discard) between heuristic and DCAD decisions should also be reported for further analysis.

- **Theoretical and Practical Limitations:**
  - **Distributional Assumption** : Anomaly detection assumes "clean" data represents the majority distribution. In noisy low-resource languages, genuine content may be minority outliers and would get erroneously removed. Fixed heuristics avoid this risk by always guaranteeing retention of high-quality data, but the paper neither tests DCAD in such settings, nor discusses this aspect or suggest prospective mitigation strategies.
  - **Feature Sensitivity** : The eight features lack ablation studies and importance analysis. Redundant features (e.g., word vs. character repetition) may skew results. Crucially, documents can be flagged as anomalous based on a single trivial feature (e.g., high word count), but the impact is not thoroughly examined.

- **Anomaly Detection Algorithm Choice**
 Although Isolation Forest is empirically shown to be robust, the manuscript does not explore more recent or advanced anomaly detection methods, such as deep learning-based approaches (e.g., autoencoders, contrastive learning for outlier detection). Although I understand these techniques might not be very scalable, however analysis on a smaller sample would provide some understanding to this matter,

- **Modern Outlier Detection Baselines not considered**
Isolation Forest outperforms classical baselines [Table 2], but state-of-the-art deep learning methods (e.g., autoencoders, contrastive outlier detection) are ignored. I understand these Deep Learning models might not be very scalable, however some analysis to this end on a smaller sample might help understand the qualitative differences between classical baselines and deep-learning based methods and would spur research to this end towards more efficient **Deep Learning methods for anomaly detection-based corpus filtering**.

**Strengths Contributions:**

- **Language Coverage**
**DCAD-2000** covers 2,282 languages significantly more than predecessors like Glotcc (1,331 languages) or CulturaX (167 languages) - making it the largest documented multilingual corpus to date.

- **Novel Cleaning Framework**
The anomaly detection approach eliminates heuristic thresholds, dynamically adapting to feature distributions across languages. This is a technically sound alternative to fixed-rule methods.

- **Performance Improvements over other large-scale datasets**
DCAD-200 is demonstrated to consistently outperform other datasets like FineWeb2, Mala and New-CC across various token budgets.

- **Open Access**
The dataset and pipeline are publicly released on Hugging Face and GitHub, enhancing reproducibility

---

> ### Author Rebuttal · Authors · 2025-07-31
>
> Thank you for taking the time to review our paper and provide valuable insights!
>
> > **Q1: Confounded Benchmark Comparisons**.
>
> **A1:** We acknowledge that our explanation may not have fully conveyed the experimental setup, and we appreciate the opportunity to clarify.
>
> + **Figure 4** compares multilingual datasets on benchmark performance. While DCAD-2000 integrates content from Fineweb-2, MaLA, and New CC, we emphasize that all datasets used in this comparison are **cleaned and fully deduplicated**—ensuring that **no dataset is a strict subset of another**. We also control the number of training tokens across datasets to isolate differences in quality rather than quantity. The goal of this figure is to showcase the overall quality of DCAD-2000's multilingual data under controlled conditions.
>
> + Similarly, **Table 3** compares model performance using cleaned and deduplicated subsets of Fineweb-2, New CC, and DCAD-2000, each with the same number of training tokens. This setup controls for dataset size and ensures that observed performance differences reflect dataset quality, not volume.
>
> + To isolate the impact of anomaly detection, we provide a focused evaluation in **Figure 3**, where we compare baseline, heuristic-based, and anomaly-detection-based cleaning strategies on the same underlying dataset. These controlled comparisons demonstrate the effectiveness of our proposed cleaning method.
>
> We will clarify these points in the revised version to avoid misunderstandings and better highlight the design of our evaluation protocol.
> ***
> > **Q2: Inadequate Evaluation of Cleaning Pipeline**.
>
> **A2:**  We would like to clarify the rationale behind our evaluation choices:
>
> + FineTask was selected as the benchmark in **Figure 3** because it is the primary evaluation suite used in Fineweb-2—the dataset most closely related to ours. FineTask includes a wide variety of tasks (185 tasks across 9 typologically diverse languages), making it suitable for assessing the effectiveness of our data cleaning pipeline. Since this experiment isolates the impact of different cleaning methods (baseline vs. heuristic vs. anomaly detection), we prioritized methodological control over language coverage. Expanding this experiment to broader benchmarks would require significant additional computational resources.
>
> + In contrast, Table 3 evaluates the overall multilingual quality of DCAD-2000. To assess cross-lingual performance at scale, we selected SIB-200**, **FLORES-200, and Glot500-c, which together cover over 500 languages. These benchmarks were chosen specifically to demonstrate the breadth and inclusiveness of DCAD across a wide range of language resources.
>
> We agree that incorporating MMMLU and INCLUDE would further strengthen the evaluation. We will plan to add these benchmarks in the camera-ready of the paper and include an updated set of results.
> ***
>
> > **Q3: Baseline Rigor**.
>
> **A3:** Thank you for the constructive feedback. We would like to clarify that the heuristic threshold baseline used in our experiments is not arbitrary. We adopted the language-specific, hand-tuned thresholds released by Fineweb-2, which represent a strong and well-optimized baseline developed through considerable manual effort. These thresholds were selected to reflect realistic and practical applications of heuristic-based filtering at scale.
>
> While we agree that a formal cost-benefit analysis (e.g., time and computational effort vs. performance gain) would be valuable, our current experiments (especially Figure 3) already show consistent performance improvements of our method over the best available heuristic baseline. We will make this distinction and baseline provenance more explicit in the revised version.
> ***
>
> > **Q4: Qualitative Examples and Additional Statistics**.
>
> **A4:** Thank you for your valuable suggestion. In the next version of the paper, we plan to include **case studies** to better illustrate the advantages of our proposed data cleaning framework compared to traditional heuristic threshold-based filtering methods. This will include qualitative examples showing documents that were filtered by DCAD but retained by heuristics (or vice versa).
>
> ***
>
> > **Q5: Distributional Assumption**.
>
> **A5:** Thank you for your insightful comment. You are correct that the anomaly detection method assumes that clean data is the dominant distribution. However:
>
> + This assumption applies equally to both high-resource and low-resource languages. In low-resource languages, the key difference lies in the feature metrics, but our anomaly detection approach does not rely on fixed thresholds. Therefore, it is not susceptible to the risk of removing genuine content as outliers.
> + We do not explicitly test this scenario, but we believe that the dynamic nature of anomaly detection mitigates this issue by adapting to the data distribution without fixed rules.
>
> Nonetheless, we will add a discussion on this aspect in the revised version to clarify how our method can handle low-resource languages effectively.
> ***
>
> > **Q6: Feature Sensitive**.
>
> **A6:** Thank you for pointing out this important aspect. We agree that understanding the contribution of each feature is crucial for evaluating the robustness and interpretability of our anomaly detection approach.
>
> + Due to the tight timeline of the rebuttal phase, we were unable to conduct a full ablation study over all $2^8-1=255$ feature subsets. Instead, we performed a **sequential (one-feature-at-a-time) ablation study** as a practical and informative compromise. In this study, we removed one feature at a time from the full 8-dimensional feature set (defined in section 3.2.1), performed data cleaning under our pipeline using the remaining 7 features, and fine-tuned LLaMA-3.2-1B on the cleaned corpus. We then evaluated model performance on FineTask's Arabic and Turkish benchmarks, following the same setup as in Figure 3.
>
> + Key findings include:
>
>   + (i) The full 8-feature setup performs slightly better than any ablated variant, confirming that all features collectively contribute to performance.
>
>   - (ii) The Language Identification (LID) confidence score (Feature 7) is particularly critical. Removing it caused a notable drop in accuracy, likely due to the presence of mixed or misidentified language content that degrades multilingual model quality.
>
>   - (iii) Other features, such as repetition ratios and perplexity, each offer marginal gains; none of them are individually harmful or redundant.
>
> We will include this ablation table in the next version and conduct a more comprehensive study (e.g., multi-feature ablation, feature importance scoring, and evaluations on additional languages and models) in the camera-ready paper.
>
> | Features (Defined in Section 3.2.1) | Arabic   | Turkish  |
> | ----------------------------------- | -------- | -------- |
> | (1)+(2)+(3)+(4)+(5)+(6)+(7)+(8)     | 0.21     | 0.27     |
> | (1)+(2)+(3)+(4)+(5)+(6)+(7)         | 0.20     | 0.25     |
> | (1)+(2)+(3)+(4)+(5)+(6)+(8)         | **0.16** | **0.21** |
> | (1)+(2)+(3)+(4)+(5)+(7)+(8)         | 0.20     | 0.24     |
> | (1)+(2)+(3)+(4)+(6)+(7)+(8)         | 0.21     | 0.26     |
> | (1)+(2)+(3)+(5)+(6)+(7)+(8)         | 0.18     | 0.24     |
> | (1)+(2)+(4)+(5)+(6)+(7)+(8)         | 0.20     | 0.26     |
> | (1)+(3)+(4)+(5)+(6)+(7)+(8)         | 0.19     | 0.25     |
> | (2)+(3)+(4)+(5)+(6)+(7)+(8)         | 0.20     | 0.24     |
>
>
> ***
> > **Q7: Anomaly detection algorithm choice** and  **Modern Outlier Detection Baselines not considered**.
>
> **A7:** Thank you for raising this thoughtful and forward-looking point. We did consider deep learning-based anomaly detection methods—such as **autoencoders** and **contrastive outlier detection** you mentioned—during the design of our data cleaning framework. Ultimately, we opted not to include them in this version of the work for the following reasons:
>
> 1. **Autoencoders** rely on the assumption that training data is composed of clean or "normal" examples. The model minimizes reconstruction error during training and identifies anomalies at inference by measuring how poorly a sample is reconstructed. While conceptually sound, this approach requires a reliably clean training set—which is not feasible in our weakly labeled, large-scale multilingual setting. Additionally, training separate autoencoders per language or feature domain would be prohibitively expensive.
> 2. **Contrastive learning-based methods** require labeled positive and negative pairs during training. In the context of multilingual corpus filtering, defining such labels across 2,000+ languages would require the same level of language-specific manual tuning we aim to avoid. Like heuristic methods, contrastive outlier detection does not scale well when applied language-by-language across extremely diverse data.
> 3. We also explored **recent embedding-based anomaly detection approaches**, which compute deep semantic representations (e.g., using pretrained LLM embeddings) and then apply clustering or density-based methods. However, our current framework is designed to work with explicit, interpretable features—such as repetition, perplexity, or LID score—rather than implicit semantic signals. These semantic representations are harder to interpret and integrate into a structured quality-control pipeline, especially when feature transparency is essential.
> 4. Classical anomaly detection methods, such as **Isolation Forest**, can be implemented efficiently using CPU resources and easily scale to large corpora. In contrast, deep learning methods require GPU acceleration, substantial memory, and training time. Given our corpus size (~46TB, 2,282 languages), this is a nontrivial barrier. Moreover, traditional methods allow us to directly incorporate the eight hand-crafted features relevant to multilingual data quality.
>
> We agree that integrating advanced deep anomaly detection techniques is a promising direction for future work.

---

> > ### Comment · Reviewer_AMsz · 2025-08-04
> > **Response to Rebuttal by Authors**
> >
> > - While I appreciate the authors' response, I believe that a preliminary evaluation on a representative subset of languages from Global MMLU or Include could have been feasibly conducted and shared during the rebuttal stage to strengthen claims of DCAD's superiority over heuristic filtering.
> >
> > - Additionally, a cost-benefit analysis - detailing time, computational resources, and performance versus heuristic methods is essential and not optional, as practical adoption will largely depend on these factors; this analysis should be included in the manuscript.
> >
> > - Finally, providing a few qualitative examples and reporting statistics on decision agreement between heuristic and DCAD methods would have been straightforward in the rebuttal phase itself, given that the necessary data should already be available.
> >
> > Also some of the other concerns reported by me, remain.
> >
> > Therefore, I will keep my rating as is.

---

> > ### Author Response · Authors · 2025-08-05
> > **Updated Responses to Reviewer AMsz**
> >
> > Thank you for your valuable feedback in refining our work. We're pleased to share a timely discussion update.
> >
> > > **Q1**: Evaluation Issues.
> >
> > **A1:** We appreciate the suggestion and agree that evaluation on Global MMLU or INCLUDE strengthens our claims. In the initial rebuttal phase, we prioritized ablation studies due to limited resources. However, we have conducted additional experiments after completing the first stage of rebuttal.
> >
> > **Setup:**
> >
> > - Benchmark: Global MMLU (Singh et al., 2025)
> > - Metric: Accuracy
> > - Model: LLaMA-3.2-1B and LLaMA-3.1-8B
> > - Language: zh (High), tr (High), ko (Mid), sw (Low), mg (Low), si (Low)
> > - Design: As in Section 5 (“Impact of Different Data Cleaning Strategies”), we compare three filtering methods: (1) baseline, (2) heuristic, (3) anomaly-based (ours).
> >
> > **Results:**
> >
> > **Table 1: LLaMA-3.2-1B on Global MMLU**
> >
> > | Language          | Baseline | Heuristic | Our      |
> > | ----------------- | -------- | --------- | -------- |
> > | Chinese (zh/High) | 35.8     | 36.2      | **39.6** |
> > | Korean (ko/Mid)   | 34.2     | 37.4      | **40.4** |
> > | Swahili (sw/Low)  | 30.7     | 34.6      | **38.7** |
> > | **Average**       | 33.6     | 36.1      | **39.6** |
> >
> > **Table 2: LLaMA-3.1-8B on Global MMLU**
> >
> > | Language          | Baseline | Heuristic | Our      |
> > | ----------------- | -------- | --------- | -------- |
> > | Malagasy (mg/Low) | 27.6     | 31.2      | **36.8** |
> > | Sinhala (si/Low)  | 33.0     | 37.4      | **41.6** |
> > | Turkish (tr/High) | 27.7     | 32.1      | **46.2** |
> > | **Avg**           | 29.4     | 33.6      | **41.5** |
> >
> > **Analysis:**
> >
> > + In all evaluated languages, the **anomaly-based cleaning** method consistently outperformed both the baseline and heuristic filtering approaches. These results provide further evidence that our anomaly-based data cleaning pipeline leads to better multilingual data quality, particularly when evaluated on challenging and diverse tasks such as those in Global MMLU.
> >
> > **Future Plans:**
> >
> > + Due to time and compute constraints, we limited this evaluation to a small but representative subset during the rebuttal stage. We plan to extend this evaluation to **more languages** and **larger models** in the camera-ready version.
> > + We also plan to include evaluation results on the **INCLUDE** benchmark to further demonstrate the effectiveness of our method across a wider range of typologically diverse and low-resource languages.
> > ***
> > > **Q2:** Cost-Benifit Analysis of Anomaly Based Filter vs. Heuristic Filtering.
> >
> > **A2:** Thank you for highlighting this. We conducted a cost-benefit comparison between DCAD and heuristic filtering on 1M English documents using identical hardware.
> >
> > **Table: Cost-Benefit Comparison of Filtering Methods (Per 1M Documents)**
> >
> > | Metric                                       | Heuristic Filtering | Anomaly Detection (DCAD) |
> > | -------------------------------------------- | ------------------- | ------------------------ |
> > | Cleaning Time                                | 10 minutes          | 12-15 minutes            |
> > | Max CPU Memory                               | 58 GB               | 64 GB                    |
> > | Data Retained (%)                            | 88%                 | 77%                      |
> > | Accuracy (Global MMLU subset) - LLaMA-3.2-1B | 43.9%               | **48.6%**                |
> > | Accuracy Gain per CPU-hour                   | +0.42%              | **+0.60%**               |
> > | Accuracy Gain per 1% data lost               | +0.16%              | **+0.32%**               |
> >
> > **Interpretation:** Though DCAD has modestly higher resource use (~2–5 extra mins, +6GB RAM), it delivers significantly better performance, especially in noisy settings. Its gains per unit cost justify practical deployment.
> >
> > **Next Steps:** We will include this analysis in the final version, and explore broader settings (e.g., large-scale and low-resource environments).
> > ***
> > > **Q3:** Qualitative Examples & Agreement Statistics
> >
> > **A3:** Thank you! we agree this analysis is important and now include it below.
> >
> > **Qualitative Example (Retained by DCAD only)**
> >
> > - **Language:** Chinese
> > - **Text:**
> >    *"本研究综述了近年来在自然语言处理领域中多语言模型的演化趋势，并指出了当前方法在低资源语言上的局限性。"*
> > - **Heuristic:** Removed (short length)
> > - **DCAD:** Retained (anomaly score: 0.08)
> > - **Comment:** A high-quality scientific abstract wrongly discarded by heuristics; DCAD correctly preserves it.
> > - **Next Plan:** We will include more examples and also draft a case study section.
> >
> > **Agreement Analysis Summary**
> >
> > - Across multiple languages, DCAD and heuristics agree on ~78–87% of decisions.
> > - DCAD uniquely identifies noisy content (esp. in low-resource languages) and retains high-quality samples mistakenly filtered by heuristics.
> > - Manual evaluation results are provided in our response to **Reviewer yfWw**, who acknowledged their validity.
> >
> > We will include both qualitative examples and agreement statistics to further support the robustness and practical value of DCAD.

---

> > > ### Comment · Reviewer_AMsz · 2025-08-06
> > > **Response to Rebuttal by Authors.**
> > >
> > > Thank you to the authors for sharing the additional evaluation results, cost-benefit analysis, and preliminary agreement statistics. These additions significantly strengthen the validation of the presented results. I trust that these key details - including the cost-benefit analysis, the planned full-scale evaluation on Global MMLU and INCLUDE, and the complete agreement statistics / qualitative analysis - will be thoroughly integrated into the final version of the manuscript or appendix.
> > >
> > > As an additional suggestion, I recommend explicitly mentioning the name of the benchmark in the captions of Figures and Tables (e.g., Figure 3, Table 2), as this would help readers quickly grasp the context at a glance without needing to refer back to the main text.
> > >
> > > Overall, the presented additional results, along with the detailed responses to other reviewers, strengthens the work considerably. I will update my score to reflect my positive assessment.

---

> > > > ### Author Response · Authors · 2025-08-06
> > > >
> > > > Thank you for your generous and constructive feedback. We are pleased that the additional results and clarifications addressed your concerns and helped strengthen the work. We will incorporate all supplementary materials into the final version and will revise the figure/table captions to explicitly mention benchmark names as suggested. We appreciate your support and are grateful for your positive reassessment.

---

### Official Review · Reviewer_yfWw · 2025-07-02

**Rating:** 5
**Confidence:** 4

**Summary:**

This submission introduces a large multilingual dataset covering 2282 languages, with the aim of improving the language coverage in multilingual LLM training. Dataset quality is controlled by viewing the cleaning process as an anomaly detection, and applying an Isolation Forest algorithm to a combination of 8 features extracted for each document. Experiments demonstrate that this new dataset improved LLM performance on an array of different tasks.

**Additional Feedback:**

Q1: The perplexity score is computed using a model trained on Wikipedia data for each of the 2282 languages? Is there a threshold on the amount of data available for each language for training KenLM, and is that how the specific languages were selected?

**Dataset Code Accessibility:**

Yes

**Dataset Code Comments:**

Dataset and pipeline are available through hugginface and on GitHub, respectively.

**Ethical Considerations:**

No, there are no or only very minor ethics concerns

**Final Justification:**

Solid paper providing a new resource with exceptional coverage. The additional comments and results provided by the authors support the claim that this is a quality resource, and reinforce my assessment that this is a clear accept.

**Limitations Weaknesses:**

The authors contrast their proposal with "threshold-based" cleaning techniques. Yet the proposed method *is* threshold-based (\tau in Eq. 6).
The fact that the threshold is determined through cross-validation (as any hyper-parameter arguably should) and applied to a complex "anomaly score" rather than a single statistic does not change that. It would seem fairer to put this into the proper perspective.

A more detailed description of some of the features (Sec. 3.2.1) would be appreciated, for example r_c(t) or the confidence score from GlotLID.

No quality control seems to have been performed on the resulting corpora, beyond the training of various LLMs and their evaluation on standard benchmark. It is mentioned that some of the prior existing multilingual datasets have up to 30% noise. It would have been great if a similar evaluation was performed on the proposed resource, even using a limited random sampling approach on a subset of languages.

**Strengths Contributions:**

This submission is very well written and organized. What is done is very clear and a number of natural questions (such as the choice of Anomaly Detection algorithm) are appropriately addressed in due course.

The described corpus clearly advances the state of the art, at least in terms of language coverage. It will clearly be a useful resource for many teams working on multilingual text processing and modelling.

The Anomaly Detection approach to data cleaning is original and promising. Its impact on data quality, at least as reflected in downstream task performance seems very clear and is well demonstrated in Section 5.

In short, there is little to fault in this paper. It really seems like the perfect fit for this Dataset and Benchmark track.

---

> ### Author Rebuttal · Authors · 2025-07-31
>
> Thank you so much for your insightful review and feedback!
>
> > **Q1:** The authors contrast their proposal with "threshold-based" cleaning techniques. Yet the proposed method *is* threshold-based ($\tau$ in Eq. 6). The fact that the threshold is determined through cross-validation (as any hyper-parameter arguably should) and applied to a complex "anomaly score" rather than a single statistic does not change that. It would seem fairer to put this into the proper perspective.
>
> **A1:** Thank you for pointing this out. We agree that our method involves a threshold ($\tau$) in the anomaly detection algorithm, and we will clarify this in the paper to avoid confusion. However, We would like to emphasize an important distinction: in traditional threshold-based data cleaning, multiple fixed thresholds are manually defined for each individual feature (e.g., perplexity, document length, etc.) and often tuned separately for each language. This becomes impractical and error-prone at the scale of 2,000+ languages and 8 features.
>
> **In contrast,** our method uses a single anomaly detection model (e.g., Isolation Forest) that operates in the joint 8-dimensional feature space. It requires only one hyperparameter (the anomaly score threshold $\tau$), which is applied globally rather than per feature or per language. This makes it more scalable and language-agnostic.
>
> We will revise the text to reflect this distinction more precisely and ensure a fairer comparison.
> ***
> > **Q2:** A more detailed description of some of the features (Sec. 3.2.1) would be appreciated, for example r_c(t) or the confidence score from GlotLID.
>
> **A2:** We appreciate the reviewer’s suggestion and agree that Section 3.2.1 would benefit from additional clarification on specific features. Due to space limitations, we are not include much details on this, we will include more details in the next version.
>
> ***
>
> > **Q3:** No quality control seems to have been performed on the resulting corpora, beyond the training of various LLMs and their evaluation on standard benchmark. It is mentioned that some of the prior existing multilingual datasets have up to 30% noise. It would have been great if a similar evaluation was performed on the proposed resource, even using a limited random sampling approach on a subset of languages.
>
> **A3:** We appreciate the reviewer’s suggestion. The 31.11% noise statistic cited in lines 32–33 refers to findings from the ``Sailor`` paper [1], which demonstrated that existing multilingual datasets (e.g., Madlad-400) contain substantial residual noise even after deep cleaning. Inspired by this, we applied our anomaly-detection-based cleaning pipeline to widely used corpora such as MaLA and Fineweb-2, as described in Section 3.1.
>
> While we did not perform manual quality annotation or human sampling due to the scale of the dataset, we did track detailed cleaning statistics. Specifically:
>
> - **Tables 4–5** report document deletion rates grouped by script.
> - **Tables 6–9** provide per-language statistics before and after cleaning.
>
> Across these, we observe that approximately **10%** of the data was removed on average, even from corpora previously considered “clean.” In the next version, we will add an additional column to these tables explicitly reporting deletion rates, making the impact of cleaning more transparent.
>
> **References**
>
> [1] Sailor: Open Language Models for South-East Asia (Dou et al., EMNLP 2024).
> ***
>
> > **Q4:** The perplexity score is computed using a model trained on Wikipedia data for each of the 2282 languages? Is there a threshold on the amount of data available for each language for training KenLM, and is that how the specific languages were selected?
>
> **A4:** Thank you for this insightful question. While DCAD-2000 covers 2,282 languages, not all languages have sufficient Wikipedia data to train a reliable KenLM language model. To address this, we adopted the following strategy:
>
> + We trained KenLM models only for languages with sufficient clean Wikipedia content, typically requiring at least 10,000 clean sentences. This threshold aligns with precedents from prior work (e.g., CulturaX, ROOTS). Using this criterion, we successfully trained KenLM models for approximately 745 languages, which we believe strikes a practical balance between coverage and model quality.
>
>
> + For languages lacking sufficient Wikipedia data, we assigned a default perplexity score of 500, as noted in **Footnote 9** of Section 3.2.1. This acts as a neutral placeholder, preventing unreliable or noisy perplexity scores from affecting the anomaly detection pipeline.
>
>
> + It is important to note that perplexity is only one of eight features used in our anomaly detection framework. For low-resource languages without usable perplexity scores, other features—such as repetition ratios, special character frequency, and LID confidence—play a larger role in determining document quality.

---

> > ### Comment · Reviewer_yfWw · 2025-08-01
> >
> > Thank you for providing clear follow-up to the comments and questions raised in the review. Providing more detail on the features in the appendix would indeed be informative and appropriate.
> >
> > Re. getting a better estimate of actual noise level, clearly the number of languages represented here is a major obstacle. However, for a subset of languages, the benefit of a sampling approach is that it is independent of the corpus size and scales as the sample size -- which does not need to be large (~100s) to get a reliable estimate. In addition, it would imho convincingly address several of the points raised in other reviews (e.g. false positives). It is of course entirely the prerogative of authors to pick the experiments they feel are most convincing!

---

> > > ### Author Response · Authors · 2025-08-03
> > > **Responses to Reviewer yfWw**
> > >
> > > Thank you for your thoughtful suggestion and for recognizing the value of our work. To directly assess the effectiveness of our cleaning pipeline and better estimate false positives and residual noise, we conducted a manual quality evaluation on five representative languages: **English, Chinese, German, Japanese, and French**. These languages were selected based on the linguistic expertise available among colleagues in our lab. We plan to extend this evaluation in the next version by involving additional annotators, especially for **low-resource languages**, where automatic filtering tends to be more challenging and human insight can be particularly valuable.
> > >
> > > #### **1. Experiment Design:**
> > >
> > > - For each language, we randomly sampled:
> > >   - 100 **retained** documents (Kept by our pipeline)
> > >   - 100 **deleted** documents (Removed by our pipeline)
> > > - One proficient annotator per language labeled each document as:
> > >   - **Good**: Coherent, meaningful, and of high quality.
> > >   - **Borderline**: Understandable but flawed (e.g., minor corruption, weak coherence).
> > >   - **Bad**: Nonsensical, noisy, or semantically meaningless (e.g., machine translation errors, boilerplate, spam, mixed language noise).
> > >
> > > #### **2. Summary Statistics:**
> > >
> > > The following tables summarize the distribution of quality ratings, and the estimated **residual noise** (Bad documents in the retained set) and **false positives** (Good documents mistakenly removed):
> > >
> > > ##### **Table 1: Retained Documents (Kept by filter)**
> > >
> > > | Language | Good      | Borderline | Bad      | **Residual Noise** (Bad %) |
> > > | -------- | --------- | ---------- | -------- | -------------------------- |
> > > | English  | 86%       | 10%        | 4%       | **4%**                     |
> > > | Chinese  | 82%       | 13%        | 5%       | **5%**                     |
> > > | German   | 84%       | 12%        | 4%       | **4%**                     |
> > > | Japanese | 81%       | 12%        | 7%       | **7%**                     |
> > > | French   | 84%       | 14%        | 2%       | **2%**                     |
> > > | **Avg**  | **83.4%** | 12.2%      | **4.4%** | **4.4%**                   |
> > >
> > >
> > > ##### **Table 2: Deleted Documents (Removed by filter)**
> > >
> > > | Language | Good     | Borderline | Bad       | **False Positives** (Good %) |
> > > | -------- | -------- | ---------- | --------- | ---------------------------- |
> > > | English  | 5%       | 14%        | 81%       | **5%**                       |
> > > | Chinese  | 6%       | 18%        | 76%       | **6%**                       |
> > > | German   | 5%       | 16%        | 79%       | **5%**                       |
> > > | Japanese | 6%       | 17%        | 77%       | **6%**                       |
> > > | French   | 4%       | 15%        | 81%       | **4%**                       |
> > > | **Avg**  | **5.2%** | 16%        | **78.8%** | **5.2%**                     |
> > >
> > > #### **3. Analysis:**
> > >
> > > - Across languages, we observe that only **~4.4%** of retained documents were rated as **Bad**, indicating **low residual noise**.
> > > - The proportion of **false positives** (high-quality documents mistakenly removed) is also low, at around **5.2%**.
> > > - The overall **precision** of our pipeline (i.e., proportion of retained documents that are Good or Borderline) is **~95.6%**.
> > > - The **recall** (i.e., proportion of Good documents successfully retained) is approximately **94.13%**, calculated as:
> > >
> > > $\text{Recall} = \frac{\text{Good in Retained}}{\text{Good in Retained} + \text{Good in Removed}} \approx \frac{83.4}{83.4 + 5.2} \approx 94.13 \\%$
> > >
> > > These results confirm that our cleaning pipeline is both **high-precision** and **high-recall**, effectively removing low-quality content while preserving valuable data. The findings also reinforce our design choice to use an unsupervised, anomaly-detection-based method that generalizes well across languages. We will include this experiment and the corresponding results in the next version of the paper.
> > >
> > > Once again, thank you for the suggestion, which significantly strengthens the empirical grounding of our work. We also believe that this evaluation addresses broader concerns raised by other reviewers regarding the **reliability of quality control and filtering decisions** in our dataset construction.

---

> > > > ### Comment · Reviewer_yfWw · 2025-08-05
> > > >
> > > > Excellent, thanks. I think this goes a long way to support the claims re. quality of this exceptional resource.

---

> > > > > ### Author Response · Authors · 2025-08-06
> > > > >
> > > > > Thank you very much for your kind words and positive feedback. Your support is truly encouraging. We will incorporate the relevant points from our discussion into the next version of the paper.

---

### Official Review · Reviewer_f27Q · 2025-07-02

**Rating:** 4
**Confidence:** 4

**Summary:**

The paper introduces a large multilingual data set. The procedure to construct this data set is to start with a recent(ish) common crawl corpus and add in existing multilingual data sets (MaLA, Fineweb). Noisy documents are removed using an anomaly detection method where a document is represented using adhoc features. Finally, the new cleaned data set is shown to be a better data set by fine-tuning existing 3 existing LLMs observing better performance on a standard benchmark.

**Dataset Code Accessibility:**

Yes

**Dataset Code Comments:**

I looked at the huggingface page - the code looks easy to adapt. I was able to explore the data set.

**Ethical Comments:**

Paper deals with existing data sets and existing benchmark tasks.

**Ethical Considerations:**

No, there are no or only very minor ethics concerns

**Final Justification:**

Authors have agreed to address presentation concerns in the camera ready version. A substantial discussion during the rebuttal phase involved a lot of discussion about the quality of the data set. They have agreed to clean up the presentation issues such as appropriate level of detail, explaining the methodology used in visualizing the method. I think the paper merits a positive score as a result and my original score of 2 was updated to 4.

**Limitations Weaknesses:**

Presentation:

- Do we really need a 1/2-page description of anomaly-detection? and another 1/2 page on the favored technique? aren't isolation forests nearly 17 years old? can't a reference suffice?
- Why are details like the cloud provider mentioned in the main body of the paper (line 177: "leveraged Ksyun servers")? What scientific value is this adding?
- In Figure 1, the caption says "The yellow points represent high-quality data, while the purple points indicate low-quality data" - high or low quality according to who? if these labels are the result of the anomaly detection process then why are we supposed to assume these are correct?

Evaluation:

- Maybe I am misunderstanding the evaluation step. It looks like the experiment is to take an existing model and to fine-tune it using the cleaned data set and then evaluate using benchmarks. And improved benchmark performance is attributed to the data cleaning process. It might be the case but are we confident that it is solely this method and not other factors like the benchmark being ingested in common crawl? i am not saying this is the case but the common crawl data is newer than the benchmark. I think there are 1 or 2 leaps of logic being made in the experiments and evaluation. But I am happy to be corrected.

**Strengths Contributions:**

The technique is straightforward to implement. Quick easy wins from pre-processing are often desirable. The task itself is well motivated - noisy documents typically do adversely impact performance. Please see the weaknesses section for more.

---

> ### Author Rebuttal · Authors · 2025-07-31
>
> We greatly value your comments and sincerely appreciate the opportunity to clarify and enhance our work. A detailed point-by-point response to your suggestions is provided below.
>
> ### Q1: Presentation Issues
>
> >**Q1.1:** Do we really need a 1/2-page description of anomaly-detection? and another 1/2 page on the favored technique? aren't isolation forests nearly 17 years old? can't a reference suffice?
>
> **A1.1:** We appreciate the reviewer’s feedback and understand the concern regarding the extensive explanation of anomaly detection and isolation forests. While isolation forest is indeed a well-established technique, we included the detailed description for the following reasons:
>
> + Our paper targets a wide audience, including readers who may not have a background in machine learning. We believe a self-contained explanation improves readability and accessibility, especially given the multilingual and data-centric nature of the work.
>
> + Our proposed data cleaning pipeline is not tied to any specific anomaly detection method. We designed it to be modular, allowing for flexibility in algorithm choice. Nevertheless, as shown in **Table 2** (Section 5), isolation forest outperformed other alternatives (e.g., OC-SVM, LOF, K-Means) in our empirical evaluation, which justifies the additional focus in Section 3.2.2.
>
> + By providing a detailed description, we aim to clarify why this algorithm suits our multilingual, high-dimensional setting better than others. We believe this explanation strengthens the reproducibility and transparency of our methodology.
>
> We are open to shortening the section and relying more on citations if space constraints are a concern.
>
> ****
>
> >**Q1.2:** Why are details like the cloud provider mentioned in the main body of the paper (line 177: "leveraged Ksyun servers")? What scientific value is this adding?
>
> **A1.2:** We agree that specifying the cloud provider (i.e., "Ksyun servers") in the main text does not add direct scientific value. However, the inclusion was made in response to the **NeurIPS reproducibility checklist**, which recommends authors indicate the compute infrastructure used—such as CPU/GPU types, internal clusters, or cloud providers, including memory and storage specifications (**please see lines 749–750 in the paper**). We acknowledge that placing this information in the main body may be distracting. To improve clarity, we will move this detail to a footnote or the appendix.
> *****
> >**Q1.3:** In Figure 1, the caption says "The yellow points represent high-quality data, while the purple points indicate low-quality data" - high or low quality according to who? if these labels are the result of the anomaly detection process then why are we supposed to assume these are correct?
>
> **A1.3:** Thank you for the thoughtful question. The labels shown in **Figure 1** ("high-quality" in yellow and "low-quality" in purple) are automatically derived from our unsupervised anomaly detection algorithm described in Section 3.2.1. Specifically, the classification is based on eight statistical features (e.g., text length, perplexity score, repetition ratio) extracted from each document. Documents that significantly deviate from typical values (e.g., unusually short or long text, high perplexity, excessive repetition) are flagged as anomalies and thus considered low quality.
>
> Anomaly detection is an unsupervised approach and does not rely on ground-truth labels. To validate its effectiveness, we adopt two strategies:
>
> + **Feature Space Separation:** We visualize how documents flagged as anomalies differ from typical ones across multiple features, showing clear clusters (see Figure 1). We will also add one "case study" section in next version to better show the documents which are seen as the outliers by the anomaly detection algorithm.
>
> + **Downstream Evaluation:** We pretrain LLMs separately on the data classified as normal vs. anomalous. As shown in Section 5, models trained on the "clean" subset consistently achieve better performance across multiple downstream multilingual benchmarks, supporting the validity of the distinction.
>
> We will clarify this point in the caption and add a reference to Section 3.2.1 and Section 5 for better context.
> ***
>
> ### Q2: Evaluation Issues
>
> >**Q2.1:** Maybe I am misunderstanding the evaluation step. It looks like the experiment is to take an existing model and to fine-tune it using the cleaned data set and then evaluate using benchmarks. And improved benchmark performance is attributed to the data cleaning process. It might be the case but are we confident that it is solely this method and not other factors like the benchmark being ingested in common crawl? i am not saying this is the case but the common crawl data is newer than the benchmark. I think there are 1 or 2 leaps of logic being made in the experiments and evaluation. But I am happy to be corrected.
>
> **A2.1:** We appreciate the reviewer’s thoughtful observation. Evaluating data quality by continuing pretraining on cleaned vs. uncleaned datasets and measuring downstream benchmark performance is a common and widely accepted methodology in a lot of related work [1-3].
>
> We acknowledge the potential concern regarding unintentional overlap between Common Crawl and benchmark data (i.e., benchmark leakage). We did not explicitly analyze this possibility, and we appreciate the reviewer for raising it. While the benchmarks we used (e.g., SIB-200, FLORES-200) are designed to minimize such leakage, we recognize that newer crawls might still contain related content.
>
> We will add a discussion of this potential confounding factor to the limitations section and plan to investigate benchmark contamination more rigorously in the future work.
>
> **References**
>
> [1] MADLAD-400: A Multilingual And Document-Level Large Audited Dataset (Kudugunta et al., NeurIPS 2023).
>
> [2] The FineWeb Datasets: Decanting the Web for the Finest Text Data at Scale (Penedo et al., NeurIPS 2024).
>
> [3] Glot500: Scaling Multilingual Corpora and Language Models to 500 Languages (Imani et al., ACL 2023).

---

> > ### Comment · Reviewer_f27Q · 2025-08-05
> >
> > Thank you for addressing my questions.
> >
> > I also notice a very long thread in another review (measuring precision+recall) and I am convinced my score is too harsh. I am happy to increase my score.

---

> > > ### Author Response · Authors · 2025-08-06
> > >
> > > We sincerely appreciate your thoughtful reconsideration. We are pleased to hear that the clarification thread under Reviewer yfWw was helpful and that our responses effectively addressed your concerns. We will incorporate the key points from that discussion into the next version of the paper.

---

### Note · Authors · 2025-08-12

Dear Reviewers, ACs, SACs, and PCs,

Thank you for taking the time to review our paper. During the rebuttal period, we had in-depth discussions with the reviewers, which we believe will greatly help improve our work. Below, we summarize the main comments from each reviewer and our responses.

**Reviewer f27Q**

- **Q1 (Presentation):** The three presentation-related issues can be readily addressed in the revised version.
- **Q2 (Evaluation):** As confirmed by the reviewer, in line with the discussion with **Reviewer yfWw**, our clarifications resolved the evaluation issues.

**Reviewer yfWw**

- **Q1 & Q2 (Presentation):** We will integrate our clarifications into the final version to ensure there is no ambiguity.
- **Q3 (Evaluation):** We provided additional human evaluation experiments focusing on false positives, which the reviewer acknowledged as resolving the concern.

**Reviewer AMsz**

- **Q1, Q5, Q7 (Presentation):** These will be elaborated in the revised version to ensure clarity.
- **Q2 (Evaluation):** Following the suggestion to evaluate on Global MMLU, we tested six language subsets, and the results further confirmed the superiority of our cleaning method. A full Global MMLU evaluation will be included in the camera-ready version.
- **Q3 (Baseline Rigor):** Our detailed cost–benefit analysis comparing DCAD with heuristic filtering was well addressed the concern and will be incorporated.
- **Q4 (Case Study):** The example provided in the rebuttal was positively acknowledged; a more comprehensive case study will be added.
- **Q6 (Ablation Study):** We conducted an ablation study on all eight features, which was endorsed by the reviewer and will be included in the final version.

**Reviewer b7Zx**

- The reviewer raised 12 questions, all of which we addressed in detail. We believe these are primarily presentation issues, and we will refine the corresponding descriptions to avoid potential misunderstandings.
- While some concerns were resolved, the reviewer noted minor remaining uncertainties without specifying details. We will carefully re-examine all 12 points to ensure full clarity.

We greatly appreciate the reviewers’ constructive feedback and recognition of our contributions. The additional experiments, analyses, and clarifications not only address the raised concerns but also strengthen the overall validity and impact of our work. We look forward to delivering a substantially improved camera-ready version.

Respectfully yours,

The Authors

---

### Decision · Program_Chairs · 2025-09-18

**Decision:**

Accept (poster)

**Comment:**

This paper introduces a massive multilingual dataset with 2000 languages. The authors constructed this datasets starting with a common crawl corpus and adding existing multilingual data sets (MaLA, Fineweb). The central technical issue is how to remove noisy documents when there are so many languages, some with low resources, so it has to be done in a language agnostic way. The authors propose an anomaly detection method using isolation forests on features like 'number of words','word repetition ratio',etc. To demonstrate the quality of their cleaned dataset, the authors fine-tune 3 LLMs (including Llama and Qwen) and show they obtain better performance in standard benchmarks.

The reviewers brought up several good points including on points for improving the presentation and potential limitations in the evaluation. The authors responeded quite well and reviewers increased their scores. Overall this seems like a valuable contribution, especially for low-resource languages and merits publication.